# Satori: Reinforcement Learning with Chain-of-Action-Thought Enhances LLM Reasoning via Autoregressive Search

**Maohao Shen** [* 1]  **Guangtao Zeng** [* 2]  **Zhenting Qi** [* 3]
**Zhang-Wei Hong** [1]  **Zhenfang Chen** [4]  **Wei Lu** [2]  **Gregory Wornell** [1]
**Subhro Das** [4]  **David Cox** [4]  **Chuang Gan** [4 5]

## Abstract

Large language models (LLMs) have demonstrated remarkable reasoning capabilities across diverse domains. Recent studies have shown that increasing test-time computation enhances LLMs' reasoning capabilities. This typically involves extensive sampling at inference time guided by an external LLM verifier, resulting in a two-player system. Despite external guidance, the effectiveness of this system demonstrates the potential of a single LLM to tackle complex tasks. Thus, we pose a new research problem: *Can we internalize the searching capabilities to fundamentally enhance the reasoning abilities of a single LLM?* This work explores an orthogonal direction focusing on post-training LLMs for autoregressive searching (*i.e.,* an extended reasoning process with self-reflection and self-exploration of new strategies). To achieve this, we propose the Chain-of-Action-Thought (COAT) reasoning and a two-stage training paradigm: 1) a small-scale format tuning stage to internalize the COAT reasoning format and 2) a large-scale self-improvement stage leveraging reinforcement learning. Our approach results in Satori, a 7B LLM trained on open-source models and data. Extensive empirical evaluations demonstrate that Satori achieves state-of-the-art performance on mathematical reasoning benchmarks while exhibits strong generalization to out-of-domain tasks. Code, data, and models are fully open-sourced[1].

---

* Core contributors of Satori team, contributed equally to this work. Order determined by a dice roll.

[1]MIT [2]Singapore University of Technology and Design [3]Harvard [4]MIT-IBM Watson AI Lab, IBM Research [5]UMass Amherst. Correspondence to: Maohao Shen <maohao@mit.edu>.

*Proceedings of the 42nd International Conference on Machine Learning*, Vancouver, Canada. PMLR 267, 2025. Copyright 2025 by the author(s).

[1]https://satori-reasoning.github.io/

## 1. Introduction

Large language models (LLMs) have demonstrated remarkable performance across a wide range of reasoning tasks, including mathematical problems (Cobbe et al., 2021; Hendrycks et al., 2021a), programming (Chen et al., 2021; Zhuo et al., 2024) and logical reasoning (Han et al., 2024; Liu et al., 2020). One of the key techniques enabling these strong reasoning capabilities is Chain-of-Thought (CoT) prompting (Wei et al., 2022), which allows LLMs to address complex tasks by generating a series of intermediate reasoning steps. As a result, many early efforts focus on fine-tuning LLMs using large-scale, high-quality CoT reasoning chains, either through human annotation (Hendrycks et al., 2021a; Yue et al., 2024) or by distilling synthetic data from more advanced models (Yu et al., 2024; Toshniwal et al., 2024a; Ding et al., 2024). However, human annotation is extremely labor intensive, and distillation often limits the model's reasoning capabilities to certain level.

Apart from scaling up training resources, more recent work has focused on test-time scaling, i.e., allocating additional inference-time compute to search for more accurate solutions. This often involves extensive sampling, either by generating multiple complete solutions (Wang et al., 2023) or by sampling multiple intermediate reasoning steps (Yao et al., 2024; Wan et al., 2024). These methods typically require external feedback to guide the search process, usually through training an auxiliary reward model to rate final solutions or intermediate steps (Sun et al., 2024; Wang et al., 2024a). However, such two-player frameworks incur more model-deployment costs and do not internalize the search capabilities into a single LLM.

Orthogonal to the above work, our study investigates a new direction that enables LLMs with autoregressive search capabilities, i.e., an extended reasoning process with self-reflection and self-exploration of new strategies. Specifically, we introduce the Chain-of-Action-Thought (COAT) mechanism, which enables LLMs to take various meta-actions during problem solving. Unlike conventional post-training consisting of large-scale supervised fine-tuning (SFT) and reinforcement learning from human feedback

(RLHF), we propose a novel two-stage training paradigm: (1) a small-scale format tuning (FT) stage to internalize the COAT reasoning format and (2) a large-scale self-improvement stage that utilizes reinforcement learning with "Restart and Explore" (RAE) techniques. Our approach leads to the development of Satori, a 7B LLM trained on open-source base models and mathematic data that achieve superior performance on both in-domain and out-of-domain tasks. To summarize, our contributions are threefold,

1. **Efficiency:** Satori is a single LLM capable of autoregressive search without external guidance (Section 6 and Section A). Moreover, this is achieved with minimal supervision and large-scale self-improvement.

2. **Effectiveness:** Satori demonstrates superior performance on in-domain mathematical reasoning tasks and outperforms the instruct model built on the same base model (Section 5.1).

3. **Generalizability:** Unlike recent research on math reasoning, Satori exhibits strong transferability to out-of-domain tasks and demonstrates universal capabilities for self-reflection and self-exploration (Section 5.2).

## 2. Related Work

We summarize the literature that is closely aligned with the scope of this paper (refer to Section B for more discussions).

**Concurrent Work.** Building on the impact of OpenAI's o1 (OpenAI, 2024), significant efforts have been made within the research community to enhance open-source LLMs with advanced reasoning capabilities. The most common approach relies on distilling knowledge from stronger teacher models (Huang et al., 2024a; Zhao et al., 2024; Min et al., 2024). In contrast, Satori addresses this problem from a reinforcement learning (RL) perspective and requires minimal supervision (only 10K samples in the format tuning stage). The most related concurrent work is DeepSeek's recently released R1 (Guo et al., 2025), which adopts a similar high-level strategy of small-scale cold-start SFT followed by large-scale RL training. Although both works coincide in this high-level idea, our work differs from R1 in key methodologies, including the data synthesis framework and RL algorithms. Additionally, DeepSeek-R1 focuses on training large-scale LLMs (671B), whereas our work provides insights into the development of smaller-scale LLMs (7B) for research purpose. Finally, as an industry-developed model, the technical details of DeepSeek-R1 (Guo et al., 2025) are not fully disclosed, making reproduction difficult, whereas our work is a fully transparent study that aims to open-source training data and training recipes.

**Post-training LLMs for Reasoning.** Recent advancements have focused on extensive post-training to enhance reasoning. A line of work focus on constructing high-quality instruction-tuning datasets (Hendrycks et al., 2021a; Yue et al., 2024; Yu et al., 2024; Toshniwal et al., 2024a; Ding et al., 2024), but suffers from expensive annotatoin costs. More recent research has focused on self-improvement approaches, where models are trained on data generated by themselves (Zelikman et al., 2022; 2024; Singh et al., 2024; Zhang et al., 2024a). Additionally, reinforcement learning methods, particularly those based on Proximal Policy Optimization (PPO) (Schulman et al., 2017a; Ouyang et al., 2022), have been demonstrated to be more effective, which typically leverage reward models to guide the learning process (Sun et al., 2024; Wang et al., 2024a; Yuan et al., 2024).

**Enabling LLMs with Searching Abilities.** Prompting-based approaches (Yao et al., 2024; Shinn et al., 2024; Hao et al., 2023; Qi et al., 2024a) guide LLMs to search for solutions via error correction and exploring alternative paths. However, such approaches cannot fundamentally enhance the LLM's reasoning abilities. Moreover, recent work has pointed out the difficulties of LLMs in self-correction (Zhang et al., 2024b; Kamoi et al., 2024). Recent research has pivoted toward training LLMs for self-exploration. Some focused on enabling *trajectory-level search*—iteratively identify errors in previous complete responses and produce improved responses (Saunders et al., 2022a; Kumar et al., 2024; Qu et al., 2024; Havrilla et al., 2024). Another line of research has explored *step-level search*, which enables LLMs to identify and correct mistakes in a more fine-grained manner. Some achieve this using another model to provide step-level feedback (Xi et al., 2024; Setlur et al., 2024; Zhang et al., 2024c; Guan et al., 2025; Zhang et al., 2024d), but such two-player frameworks suffer from high costs for model deployment. SoS (Gandhi et al., 2024) is another closely related work that attempts to train a single LLM to perform a tree search as a flattened string. However, the effectiveness of SoS has primarily been shown on simple symbolic tasks, and its ability to generalize to more complex problems remains to be explored.

## 3. Preliminaries

We address mathematical problem-solving by training a language model $\pi_\theta$ to generate a solution $\tilde{y}$ that matches the ground truth $y^*$, given a problem prompt $x$. All sequences $x$, $y$, and $y^*$ consist of tokens from a predefined dictionary. Since our approach uses reinforcement learning (RL) to train the model for solving math problems, we outline the key RL concepts below.

**Reinforcement Learning (RL).** RL (Kaelbling et al., 1996) involves an agent making sequential decisions to max-

*Figure 1:* **A High-level Overview of Satori Training Framework: Format Tuning (FT) + Self-improvement.** First, Satori learns COAT reasoning format through imitation learning on **small-scale** demonstration trajectories. Next, Satori further leverages COAT reasoning format to self-improve via **large-scale** reinforcement learning.

imize the expected cumulative rewards through interactions with an environment. Here, the language model $\pi_\theta$ acts as the agent's policy. Starting from an initial state $z_0$, at each step $l$, the agent observes the current state $z_l$, receives a reward $r_l$, selects an action based on $\pi_\theta$, transitions to the next state $z_{l+1}$, and continues until reaching a terminal state. A trajectory is the sequence of states and actions during this interaction. RL optimizes the policy to maximize the expected rewards $\sum_{l=1}^{L} r_l$, where $L$ is the trajectory length.

## 4. Method

We start this section by introducing the formulation of reasoning and how reasoning can be formulated as a sequential decision-making problem. **Goal:** We want to train LLMs to solve problems by reasoning through multiple steps rather than directly predicting the final answer. Given a problem statement $x$, the model generates a sequence of reasoning steps $\{y_1, y_2, \ldots, y_L\}$, where $y_L$ provides the final answer. However, not all intermediate steps are helpful—repeating errors does not improve accuracy. Effective reasoning requires verifying correctness, identifying mistakes, and considering alternative solutions. For instance, given $x = $ "$1 + 1 =$?", the model might initially output $y_1 = 3$, then recognize the mistake with $y_2$ (e.g., *"Wait, let me verify..."*), before correcting it to $y_3 = 2$.

**Chain-of-Action-Thought reasoning (COAT).** The key challenge is enabling the model to determine when to reflect, continue, or explore alternatives without external intervention. To enable this, we introduce special *meta-action* tokens that guide the model's reasoning process beyond standard text generation. These tokens serve as hint for the model to determine when to reassess its reasoning before proceeding.

- **Continue Reasoning** (`<|continue|>`): Encourages the model to build upon its current reasoning trajectory by generating the next intermediate step.

- **Reflect** (`<|reflect|>`): Prompts the model to pause and verify the correctness of prior reasoning steps.

- **Explore Alternative Solution** (`<|explore|>`): Signals the model to identify critical flaws in its reasoning and explore a new solution.

Each reasoning step $y_l$ is a sequence of tokens, with the starting token potentially being one of the designated meta-action tokens. We refer to this formulation as Chain-of-Action-Thought reasoning (COAT). In particular, typical Chain-of-Thought reasoning (CoT) (Wei et al., 2022) can be viewed as a special case of COAT, where each reasoning step in CoT is restricted to continuation, without explicitly incorporating other types of meta-actions.

**Learning to Reason via RL.** We formulate reasoning as a sequential decision-making problem, where reasoning is a process of constructing and refining an answer step by step. Specifically, the model $\pi_\theta$ starts with an input context $x$ (initial state $z_0$), generates a reasoning step $y_l$ (action), updates the context by appending $y_l$ (next state $z_{l+1} = z_l \oplus y_l$, where $\oplus$ denotes string concatenation), and repeats this process until it produces a final answer $y_L$. The reasoning terminates when the model signals completion (e.g., omitting EOS token). The simplest reward function can be $\mathbb{I}\{y_L = y^*\}$, evaluates whether the final answer $y_L$ matches the ground truth $y^*$. With this formulation, we could train the model to reason using RL, aiming to generate reasoning steps that maximize the expected reward. However, applying RL to reasoning presents two key challenges:

1. **Unawareness of meta-action tokens:** The model doesn't understand the purpose of special tokens and fails to recognize that encountering special meta-action tokens may require reflection or proposing alternatives.

2. **Long horizon and sparse rewards:** Reasoning requires long-term decision-making with rewards only at the end, which hinders learning effectiveness (Bellemare et al., 2016). The model must take many correct reasoning steps before receiving rewards, and failures force it to restart from the initial state (i.e., the problem statement). This makes learning difficult because training data associated with rewards is scarce, yet rewards are essential for driving RL progress.

**Overview of Proposed Method.** To address the model's initial unawareness of meta-action tokens, we introduce a warm-up "format-tuning" stage: we fine-tune a pre-trained LLM on a small dataset featuring a few demonstrated reasoning trajectories (Section 4.1). This step familiarizes the model with using and reacting to meta-action tokens. Second, to tackle the challenges of long horizons and sparse rewards, we propose a "restart and explore" (RAE) strategy, inspired by Go-explore (Ecoffet et al., 2019). Here, the model restarts from intermediate steps, including those points where previous reasoning attempts failed, allowing it to focus on correcting errors rather than starting from scratch. We also add exploration bonuses to encourage deeper reflection, further increasing opportunities for the model to arrive at correct answers (Section 4.2).

### 4.1. Format Tuning Through Imitation Learning

Training a base LLM $\pi_\theta$ to perform COAT reasoning presents a significant challenge: LLMs are typically not pre-trained on COAT reasoning data that incorporates trials and errors, necessitating a post-training stage to inject this capability. To address this, we introduce format tuning (FT), a method designed to train LLMs to emulate expert COAT trajectories through imitation learning. Imitation learning techniques (Hussein et al., 2017) are widely used in the robotics domain, where agents are trained using demonstration trajectories provided by human experts (Ross and Bagnell, 2010; Ross et al., 2011; Ho and Ermon, 2016). However, generating high-quality demonstration trajectories for LLMs is prohibitively expensive for complex tasks. To efficiently construct a demonstration trajectory dataset $\mathcal{D}_{\text{syn}} = \{(\boldsymbol{x}^{(i)}, \tilde{\boldsymbol{y}}^{(i)})\}_{i=1}^N$, we propose a multi-agent data synthesis framework that leverages three LLMs:

- **Generator:** Given an input problem, a generator $\pi_g$ generates multiple reasoning paths for a given input problem using classical CoT techniques.

- **Critic:** A critic $\pi_c$ evaluates the correctness of the reasoning paths generated by the generator, providing feedback to refine the reasoning and address suboptimal steps.

- **Reward Model:** Additionally, a reward model $\pi_r$ assigns scores to the refined reasoning paths and selects the most effective path as the final demonstration trajectory.

These three models collaborate to construct high-quality demonstration trajectories (details on the trajectory synthesis are provided in Appendix C). For this work, we adopt the simplest imitation learning approach, behavior cloning, which utilizes supervised fine-tuning to train the LLM policy on the expert COAT demonstration trajectories $\mathcal{D}_{\text{syn}}$. Notably, we observe that even a small number (10K) of COAT demonstration trajectories is sufficient for $\pi_\theta$ to effectively follow the COAT reasoning format.

### 4.2. Self-improvement via Reinforcement Learning

After format tuning, the LLM policy $\pi_\theta$ adopts the COAT reasoning style but struggles to generalize, particularly in using meta-actions for self-reflection. This limitation arises from the scarcity of demonstrations during format tuning. While collecting more demonstrations could help, it is costly and time-consuming. Instead, we explore whether the model can *self-improve* its reasoning via RL.

We start with the format-tuned LLM and train it using PPO (Schulman et al., 2017b) algorithm, a widely used RL method. In addition to training on problems $\boldsymbol{x}$ from the dataset $\mathcal{D}$, we also train the model $\pi_\theta$ to begin reasoning from partial trajectories generated by the format-tuned LLM. Since reasoning errors typically arise from minor mistakes rather than fundamental flaws, re-exploring from the start is inefficient. Instead, we allow the model to restart from intermediate steps to correct errors and finally achieve correct answers. Inspired by Go-Explore (Ecoffet et al., 2019), we introduce the *Restart and Explore (RAE)* strategy.

---

**Algorithm 1** Restart and Explore (RAE)

**input** Dataset $\mathcal{D} = \{(\boldsymbol{x}^{(i)}, \boldsymbol{y}^{*(i)})\}_{i=1}^n$; LLM policy $\pi_\theta$ after format tuning; maximum back-track steps $T$

▷ Initialize $\mathcal{D}_{\text{restart}}^+ \leftarrow \emptyset$; Initialize $\mathcal{D}_{\text{restart}}^- \leftarrow \emptyset$
**for** $i = 1, 2, \ldots, n$ **do**
    ▷ Given input problem $\boldsymbol{x}^{(i)}$, sample $\pi_\theta$ and collect multiple initial trajectories.
    ▷ Randomly select one correct trajectory $\tilde{\boldsymbol{y}}^+$ and one incorrect trajectory $\tilde{\boldsymbol{y}}^-$.
    ▷ Randomly backtrack last $t \leq T$ actions from $\tilde{\boldsymbol{y}}^+$ and $\tilde{\boldsymbol{y}}^-$.
    ▷ Obtain intermediate states at time-step $L - t$, $z_{L-t}^+ = [\boldsymbol{x}^{(i)}, \tilde{\boldsymbol{y}}_1^+, \tilde{\boldsymbol{y}}_2^+, \ldots, \tilde{\boldsymbol{y}}_{L-t}^+]$; $z_{L-t}^- = [\boldsymbol{x}^{(i)}, \tilde{\boldsymbol{y}}_1^-, \tilde{\boldsymbol{y}}_2^-, \ldots, \tilde{\boldsymbol{y}}_{L-t}^-]$.
    ▷ Add "reflect" special token to trigger self-reflection action, $z_{L-t}^+ = [\boldsymbol{x}^{(i)}, \tilde{\boldsymbol{y}}_1^+, \tilde{\boldsymbol{y}}_2^+, \ldots, \tilde{\boldsymbol{y}}_{L-t}^+, \texttt{<|reflect|>}]$; $z_{L-t}^- = [\boldsymbol{x}^{(i)}, \tilde{\boldsymbol{y}}_1^-, \tilde{\boldsymbol{y}}_2^-, \ldots, \tilde{\boldsymbol{y}}_{L-t}^-, \texttt{<|reflect|>}]$.
    ▷ Update $\mathcal{D}_{\text{restart}}^+ \leftarrow \mathcal{D}_{\text{restart}}^+ \cup z_{L-t}^+$; $\mathcal{D}_{\text{restart}}^- \leftarrow \mathcal{D}_{\text{restart}}^- \cup z_{L-t}^-$.
**end**
$\mathcal{D}_{\text{restart}} = \{\boldsymbol{x}^{(i)}\}_{i=1}^n \cup \mathcal{D}_{\text{restart}}^+ \cup \mathcal{D}_{\text{restart}}^-$.

**output** Augmented initial states dataset $\mathcal{D}_{\text{restart}}$.

---

**Initial States.** RAE trains the model to reason not only from the problem statement but also from intermediate steps sampled from past trajectories, both correct and incorrect. This enables deeper exploration without redundant recomputation. As detailed in Algorithm 1, given an input problem $x \in \mathcal{D}$, the format-tuned LLM first generates multiple reasoning trajectories. We then randomly backtrack $T \geq 0$ steps and append a reflect token $\texttt{<|reflect|>}$ to prompt the model to refine its reasoning. To encourage diverse exploration, correct and incorrect trajectories are stored separately in restart buffers ($\mathcal{D}_{\text{restart}}^+$ and $\mathcal{D}_{\text{restart}}^-$). RL training then optimizes reasoning across these buffers along with the

original problem dataset, sampling initial states from the merged dataset $\mathcal{D}_{\text{restart}}$.

Intuitively, RAE modifies the initial state distribution by starting new rollouts not only from dataset-sampled prompts but also from random partial trajectories, which encourages a more diverse initial state distribution. This technique is aligned with the insight offered in (Kakade and Langford, 2002), which argues that more uniform state coverage leads to tighter bounds of learned policy. We leave a more rigorous theoretical justification for future work.

**Reward Design.** RAE gives the model multiple opportunities to refine its reasoning, but effective reflection is key to making use of these chances. In addition to using correctness as rewards, we introduce the following bonuses rewards as hints to guide the model to reach correct answers:

- **Rule-based Reward:** Rule-based reward simply evaluates the correctness of the final answer.

$$r_{\text{rule}}(\tilde{\boldsymbol{y}}_L, \boldsymbol{y}^*) = \mathbf{1}_{\tilde{\boldsymbol{y}}_L = \boldsymbol{y}^*} - 1 \in \{-1, 0\}$$

- **Reflection Bonuses:** To reinforce self-reflection, we introduce a reflection bonus $r_{\text{bonus}}$. If the model starts from an incorrect reasoning trajectory stored in the *negative restart buffer* ($\mathcal{D}_{\text{restart}}^-$) and successfully solves the problem, it obtains a reward bonus, encouraging it to correct past mistakes. Conversely, if it starts from a correct trajectory in the *positive restart buffer* ($\mathcal{D}_{\text{restart}}^+$) but fails to solve the problem, it incurs a penalty, discouraging unnecessary revisions when it was already on the right track. Formally, the reflection bonus is defined as:

$$r_{\text{bonus}}(z, \tilde{\boldsymbol{y}}) = \begin{cases} \beta & \text{if } z \in \mathcal{D}_{\text{restart}}^- \text{ and } \tilde{\boldsymbol{y}}_L = \boldsymbol{y}^*, \\ -\beta & \text{if } z \in \mathcal{D}_{\text{restart}}^+ \text{ and } \tilde{\boldsymbol{y}}_L \neq \boldsymbol{y}^*, \\ 0 & \text{otherwise,} \end{cases}$$

where $\beta$ is a bonus scale hyperparameter.

- **Preference Bonuses:** Since correct answers are rare at initial training stage, reward signals are often too sparse for effective RL training. Even with reflection, the model may fail to generate any correct reasoning trajectories, resulting in a sparse reward problem. To mitigate this, we train an Outcome Reward Model (ORM) using a Bradley-Terry (BT) preference framework. The ORM rates reasoning trajectories, assigning higher values to correct (preferred) ones. For each problem $\boldsymbol{x} \in \mathcal{D}$, we generate multiple trajectories using $\pi_\theta$ and construct a preference dataset by pairing correct and incorrect outputs. A BT model is trained to maximize the score gap between these pairs. The ORM's output, $\sigma(r_\psi(z, \tilde{\boldsymbol{y}})) \in [0, 1]$, serves as a fine-grained reward signal, helping the model further refine its reasoning. See Appendix D.3 for details.

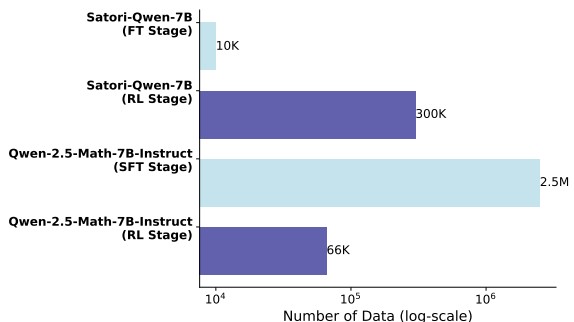

*Figure 2:* **Number of Training Samples of Satori-Qwen-7B and Qwen-2.5-Math-7B-Instruct.** Satori-Qwen-7B requires significantly less supervision (small-scale FT) and relies more on self-improvement (large-scale RL).

For an initial state $z \in \mathcal{D}_{\text{restart}}$ and a sampled trajectory $\tilde{\boldsymbol{y}}$, the overall reward function $r(z, \tilde{\boldsymbol{y}})$ is defined as:

$$r(z, \tilde{\boldsymbol{y}}) = r_{\text{rule}}(\tilde{\boldsymbol{y}}_L, \boldsymbol{y}^*) + \sigma(r_\psi(z, \tilde{\boldsymbol{y}})) + r_{\text{bonus}}(z, \tilde{\boldsymbol{y}})$$

**Iterative Self-improvement.** RL enables a policy to self-improve from self-generated trajectories, but it can also lead to a vicious cycle, where the policy converges to a local sub-optimum and cannot further improve. Inspired by (Agarwal et al., 2022; Schmitt et al., 2018), we propose an iterative self-improvement strategy to mitigate this issue. Specifically, after each round of RL training, we distill the knowledge of the current well-optimized policy into the base model through supervised fine-tuning (SFT). Starting from the newly fine-tuned model, we then perform another round of RL training. Intuitively, from an optimization perspective, each round of distillation can be viewed as a parameter reset mechanism that helps the policy escape local optima in the loss landscape, allowing it to continue self-improving (more details are included in Section D.3). In the next section, we provide empirical evidence to validate this approach.

## 5. Experiment

**Implementation Details.** We employ Qwen-2.5-Math-7B as the base model due to its strong mathematical capabilities. Our training data is sourced from the publicly available math instruction datasets, OpenMathInstruct-2 and NuminaMath-CoT. For the multi-agent data synthesis framework, the generator is required to generate high-quality, step-by-step reasoning trajectories. Therefore, we use Qwen-2.5-Math-Instruct as the generator. Meanwhile, the critic must have robust instruction-following capabilities, so we choose Llama-3.1-70B-Instruct as the critic. To ensure data quality, we filter out problems with invalid questions or incorrect labels, resulting in approximately 550k samples. Additional implementation details can be found in Appendix D.

*Table 1:* **Results on Mathematic Benchmarks.** Satori-Qwen-7B achieves SOTA performance across five benchmarks, and outperforms Qwen-2.5-Math-7B-Instruct which uses the same base model Qwen-2.5-Math-7B. After round-2 training, Satori-Qwen-7B (Round 2) demonstrates even stronger performance on hard tasks.

| Scale | Model | GSM8K | MATH500 | OlympiadBench | AMC2023 | AIME2024 | Avg. |
|-------|-------|-------|---------|---------------|---------|----------|------|
| Large | GPT-4o | / | 60.3 | 43.3 | / | 9.3 | / |
| | o1-preview | / | 85.5 | / | 82.5 | 44.6 | / |
| | Llama-3.1-70B-Instruct | 94.1 | 68.0 | 29.4 | 42.5 | 13.3 | 49.5 |
| | OpenMath2-Llama3.1-70B | 94.1 | 71.8 | 30.1 | 45.0 | 13.3 | 50.9 |
| | QwQ-32B-Preview | 95.5 | 90.6 | 61.2 | 77.5 | 50.0 | 75.0 |
| Small | Llama-3.1-8b-Instruct | 84.4 | 51.9 | 15.1 | 22.5 | 3.3 | 35.4 |
| | OpenMath2-Llama3.1-8B | 90.5 | 67.8 | 28.9 | 37.5 | 6.7 | 46.3 |
| | NuminaMath-7B-CoT | 78.9 | 54.6 | 15.9 | 20.0 | 10.0 | 35.9 |
| | Qwen-2.5-7B-Instruct | 91.6 | 75.5 | 35.5 | 52.5 | 6.7 | 52.4 |
| | Qwen-2.5-Math-7B-Instruct | 95.2 | 83.6 | 41.6 | 62.5 | 16.7 | 59.9 |
| | **Satori-Qwen-7B** | 93.2 | 85.6 | 46.6 | 67.5 | 20.0 | 62.6 |
| | **Satori-Qwen-7B (Round 2)** | 93.9 | 83.6 | 48.5 | 72.5 | 23.3 | 64.4 |

*Table 2:* **Results on Out-of-domain Benchmarks.** Trained only on math datasets, Satori-Qwen-7B exhibits strong transferability across diverse out-of-domain benchmarks and outperforms Qwen-2.5-Math-7B-Instruct by a large margin. Moreover, despite not being trained in other domains, Satori-Qwen-7B achieves performance comparable to or exceeding other small-scale general instruct models.

| Scale | Model | FOLIO | BGQA | CRUXEval | StrategyQA | TableBench | STEM | Avg. |
|-------|-------|-------|------|----------|------------|------------|------|------|
| Large | Llama-3.1-70B-Instruct | 65.0 | 58.3 | 59.6 | 88.8 | 34.2 | 61.7 | 61.3 |
| | OpenMath2-Llama3.1-70B | 68.5 | 68.7 | 35.1 | 95.6 | 46.8 | 15.1 | 55.0 |
| | QwQ-32B-Preview | 84.2 | 71.1 | 65.2 | 88.2 | 51.5 | 71.3 | 71.9 |
| Small | Llama-3.1-8b-Instruct | 63.5 | 50.3 | 38.5 | 92.2 | 32.4 | 43.4 | 53.4 |
| | OpenMath2-Llama3.1-8B | 57.1 | 49.0 | 11.1 | 84.4 | 34.2 | 10.9 | 41.1 |
| | NuminaMath-7B-CoT | 53.2 | 44.6 | 28.0 | 77.8 | 29.1 | 11.3 | 40.7 |
| | Qwen-2.5-7B-Instruct | 72.4 | 53.0 | 58.1 | 91.3 | 43.2 | 57.1 | 62.5 |
| | Qwen-2.5-Math-7B-Instruct | 68.9 | 51.3 | 28.0 | 85.3 | 36.2 | 45.2 | 52.5 |
| | **Satori-Qwen-7B** | 71.4 | 61.8 | 42.5 | 86.3 | 43.4 | 56.7 | 60.4 |
| | **Satori-Qwen-7B (Round 2)** | 72.9 | 58.5 | 41.1 | 90.4 | 44.6 | 57.4 | 60.8 |

**Benchmark and Evaluation.** We conduct the main evaluation of the models using math benchmarks to assess their problem-solving abilities, including GSM8K, MATH500 (a subset of the MATH test set (Lightman et al., 2023)), AMC2023, AIME2024, and OlympiadBench. Except for GSM8K, all other datasets feature competition-level problems. The evaluation is performed using greedy decoding without tool integration. The main metric reported is the zero-shot pass@1 accuracy, which measures the percentage of problems correctly solved on the first attempt. We also conduct additional evaluations on a wide range of benchmarks beyond the math domain to evaluate general reasoning capabilities. This includes logical reasoning (FOLIO (Han et al., 2024), BoardgameQA (BGQA) (Kazemi et al., 2024)), code reasoning (CRUXEval (Gu et al., 2024)), commonsense reasoning (StrategyQA (STGQA) (Geva et al., 2021)), tabular reasoning (TableBench (Wu et al., 2024a)), and domain-specific reasoning (STEM subsets of MMLU-Pro (Wang et al., 2024b)), including physics, chemistry, computer science, engineering, biology, and economics. For more evaluation details, please refer to Appendix D.4.

**Baseline Models.** We compare our developed model, Satori-Qwen-7B, with several industry-developed LLMs. The main comparison is between our model and Qwen-2.5-Math-7B-Instruct (Yang et al., 2024a), a math-specialized model built on the same base model (Qwen-2.5-Math-7B) as ours. Additionally, we report the performance of larger models, including o1-preview and QwQ-32B-Preview, which exhibit strong reasoning capabilities and serve as performance upper bounds.

**5.1. Main Results on Math Domain**

We present math benchmark results in Table 1, where Satori-Qwen-7B outperforms all small-scale baseline models. Notably, using Qwen-2.5-Math-7B as the base model, Satori-Qwen-7B achieves superior performance compared to Qwen-2.5-Math-7B-Instruct, despite requiring significantly less supervision (i.e., less SFT data) and relying more on self-improvement (i.e., more RL data) (see Figure 2).

## 5.2. Out-of-Domain Transferability

Although Satori-Qwen-7B is trained only on math domain datasets, we observe that it can extrapolate its reasoning capabilities to other domains. In Table 2, we evaluate Satori-Qwen-7B on a diverse set of out-of-domain benchmarks that require reasoning capabilities but are not directly related to math. Similar to the observation on the math domain, Satori demonstrates superior performance on several benchmarks, outperforming Qwen-2.5-Math-7B-Instruct. In particular, on the challenging reasoning benchmark BoardgameQA, Satori-Qwen-7B surpasses all baseline models of the same scale. These results and demo examples in Appendix A suggest that Satori has acquired general reasoning capabilities rather than simply math problem solving skills. In Section 6, we present further analysis to show that this transferability emerges as a result of large-scale reinforcement learning.

## 5.3. Results on Iterative Self-improvement

Finally, we present the results of the second-round training of Satori. As shown in Table 1 and Table 2, compared to Satori-Qwen-7B, Satori-Qwen-7B (Round 2) demonstrates continuous performance gains across most in-domain and out-of-domain benchmarks. This suggests the significant potential of iterative self-improvement to push the limit of LLM's reasoning performance.

## 6. Analysis

In this section, we provide a comprehensive analysis of Satori. First, we demonstrate that Satori effectively leverages self-reflection to seek better solutions and enhance its overall reasoning performance. Next, we observe that Satori exhibits test-scaling behavior through RL training, where it progressively acquires more tokens to improve its reasoning capabilities. Finally, we conduct ablation studies on various components of Satori's training framework. Additional results are provided in Appendix E.

*Table 3:* **COAT Training v.s. CoT Training.** Qwen-2.5-Math-7B trained with COAT reasoning format (Satori-Qwen-7B) outperforms the same base model but trained with classical CoT reasoning format (Qwen-7B-CoT)

| Model | GSM8K | MATH500 | Olym. | AMC2023 | AIME2024 |
|---|---|---|---|---|---|
| Qwen-2.5-Math-7B-Instruct | 95.2 | 83.6 | 41.6 | 62.5 | 16.7 |
| Qwen-7B-CoT (SFT+RL) | 93.1 | 84.4 | 42.7 | 60.0 | 10.0 |
| **Satori-Qwen-7B** | 93.2 | 85.6 | 46.6 | 67.5 | 20.0 |

**COAT Reasoning v.s. CoT Reasoning.** We begin by conducting an ablation study to demonstrate the benefits of COAT reasoning compared to the classical CoT reasoning. Specifically, starting from the synthesis of demonstration trajectories in the format tuning stage, we ablate the "reflect" and "explore" actions, retaining only the "continue"

actions. Next, we maintain all other training settings, including the same amount of SFT and RL data and consistent hyper-parameters. This results in a typical CoT LLM (Qwen-7B-CoT) without self-reflection or self-exploration capabilities. As shown in Table 3, the performance of Qwen-7B-CoT is suboptimal compared to Satori-Qwen-7B and fails to surpass Qwen-2.5-Math-7B-Instruct, suggesting the advantages of COAT reasoning over CoT reasoning.

*Table 4:* **Satori's Self-correction Capability.** T→F: negative self-correction; F→T: positive self-correction.

| Model | In-Domain | | | | Out-of-Domain | |
|---|---|---|---|---|---|---|
| | MATH500 | | OlympiadBench | | MMLUProSTEM | |
| | T→F | F→T | T→F | F→T | T→F | F→T |
| Satori-Qwen-7B-FT | 79.4% | 20.6% | 65.6% | 34.4% | 59.2% | 40.8% |
| **Satori-Qwen-7B** | 39.0% | 61.0% | 42.1% | 57.9% | 46.5% | 53.5% |

**Satori Exhibits Self-correction Capability.** We observe that Satori frequently engages in self-reflection during the reasoning process (see demos in Section A), which occurs in two scenarios: (1) it triggers self-reflection at intermediate reasoning steps, and (2) after completing a problem, it initiates a second attempt through self-reflection. We focus on quantitatively evaluating Satori's self-correction capability in the second scenario. Specifically, we extract responses where the final answer before self-reflection differs from the answer after self-reflection. We then quantify the percentage of responses in which Satori's self-correction is positive (i.e., the solution is corrected from incorrect to correct) or negative (i.e., the solution changes from correct to incorrect). The evaluation results on in-domain datasets (MATH500 and Olympiad) and out-of-domain datasets (MMLUPro) are presented in Table 4. First, compared to Satori-Qwen-FT which lacks the RL training stage, Satori-Qwen demonstrates a significantly stronger self-correction capability. Second, we observe that this self-correction capability extends to out-of-domain tasks (MMLUProSTEM). These results suggest that RL plays a crucial role in enhancing the model's true reasoning capabilities.

**RL Enables Satori with Test-time Scaling Behavior.** Next, we aim to explain how reinforcement learning (RL) incentivizes Satori's autoregressive search capability. First, as shown in Figure 3, we observe that Satori consistently improves policy accuracy and increases the average length of generated tokens with more RL training-time compute. This suggests that Satori learns to allocate more time to reasoning, thereby solving problems more accurately. One interesting observation is that the response length first decreases from 0 to 200 steps and then increases. Upon a closer investigation of the model response, we observe that in the early stage, our model has not yet learned self-reflection capabilities. During this stage, RL optimization may prioritize the model

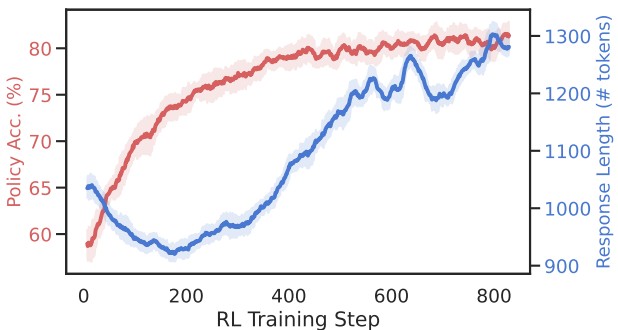

*Figure 3:* **Policy Training Acc. & Response length v.s. RL Train-time Compute.** Through RL training, Satori learns to improve its reasoning performance through longer thinking.

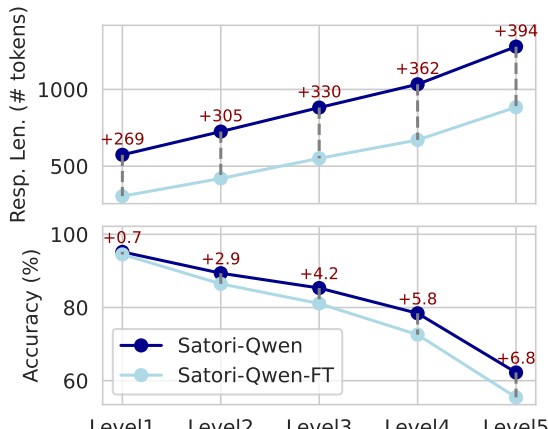

*Figure 4:* **Above: Test-time Response Length v.s. Problem Difficulty Level; Below: Test-time Accuracy v.s. Problem Difficulty Level.** Compared to FT model (Satori-Qwen-FT), Satori-Qwen uses more test-time compute to tackle more challenging problems.

to find a shot-cut solution without redundant reflection, leading to a temporary reduction in response length. However, in later stage, the model becomes increasingly good at using reflection to self-correct and find a better solution, leading to a longer response length.

Additionally, in Figure 4, we evaluate Satori's test accuracy and response length on MATH datasets across different difficulty levels. Interestingly, through RL training, Satori naturally allocates more test-time compute to tackle more challenging problems, which leads to consistent performance improvements compared to the format-tuned (FT) model.

**Large-scale FT v.s. Large-scale RL.** We investigate whether scaling up format tuning (FT) can achieve performance gains comparable to RL training. We conduct an ablation study using Qwen-2.5-Math-7B, trained with an equivalent amount of FT data (300K). As shown in Table 5, on the math domain benchmarks, the model trained with large-scale FT (300K) fails to match the performance of the model trained with small-scale FT (10K) and large-scale RL

*Table 5:* **Large-scale FT V.S. Large-scale RL** Satori-Qwen (10K FT data + 300K RL data) outperforms same base model Qwen-2.5-Math-7B trained with 300K FT data (w/o RL) across all math and out-of-domain benchmarks.

| (In-domain) | GSM8K | MATH500 | Olym. | AMC2023 | AIME2024 |
|---|---|---|---|---|---|
| Qwen-2.5-Math-7B-Instruct | 95.2 | 83.6 | 41.6 | 62.5 | 16.7 |
| Satori-Qwen-7B-FT (300K) | 92.3 | 78.2 | 40.9 | 65.0 | 16.7 |
| **Satori-Qwen-7B** | 93.2 | 85.6 | 46.6 | 67.5 | 20.0 |

| (Out-of-domain) | BGQA | CRUX | STGQA | TableBench | STEM |
|---|---|---|---|---|---|
| Qwen-2.5-Math-7B-Instruct | 51.3 | 28.0 | 85.3 | 36.3 | 45.2 |
| Satori-Qwen-7B-FT (300K) | 50.5 | 29.5 | 74.0 | 35.0 | 47.8 |
| **Satori-Qwen-7B** | 61.8 | 42.5 | 86.3 | 43.4 | 56.7 |

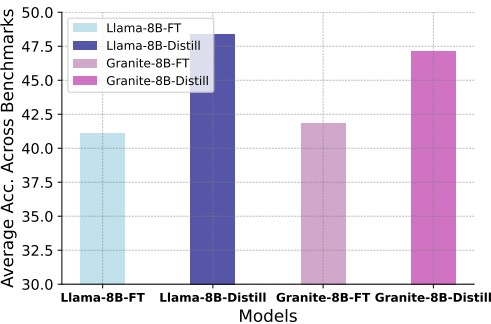

*Figure 5:* **Format Tuning v.s. Distillation.** Distilling from a Stronger model (Satori-Qwen-7B) to weaker base models (Llama-8B and Granite-8B) are more effective than directly applying format tuning on weaker base models.

(300K). Additionally, the large-scale FT model performs significantly worse on out-of-domain tasks, demonstrates RL's advantage in generalization.

**Distillation Enables Weak-to-Strong Generalization.** Finally, we investigate whether distilling a stronger reasoning model can enhance the reasoning performance of weaker base models. Specifically, we use Satori-Qwen-7B to generate 240K synthetic data to train weaker base models, Llama-3.1-8B and Granite-3.1-8B. For comparison, we also synthesize 240K FT data (following Section 4.1) to train the same models. We evaluate the average test accuracy of these models across all math benchmark datasets, with the results presented in Figure 5. The results show that the distilled models outperform the format-tuned models.

This suggests a new, efficient approach to improve the reasoning capabilities of weaker base models: (1) train a strong reasoning model through small-scale FT and large-scale RL (our Satori-Qwen-7B) and (2) distill the strong reasoning capabilities of the model into weaker base models. Since RL only requires answer labels as supervision, this approach introduces minimal costs for data synthesis, i.e., the costs induced by a multi-agent data synthesis framework or even more expensive human annotation.

## 7. Concluding Remarks

The training framework of Satori exhibits significant potential for enhancing LLM reasoning capabilities. The small-scale format tuning stage serves as a warm-up phase, allowing the LLM policy to internalize a specific reasoning format, while large-scale reinforcement learning (RL) plays a crucial role in incentivizing intrinsic reasoning abilities. We believe that this framework can inspire the research community to explore more methods for achieving autoregressive search, such as developing reasoning formats with a broader range of meta-actions, designing more advanced RL algorithms, and extending this approach to general domain.

## Acknowledgment

This research/project was supported, in part, by the following funding resources: MIT-IBM Watson AI Lab under Agreement No. W1771646, the National Research Foundation, Singapore under its AI Singapore Programme (AISG Award No: AISG2-PhD-2021-08-007), and Ministry of Education, Singapore, under its Academic Research Fund (AcRF) Tier 2 Programme (MOE AcRF Tier 2 Award No. : MOE-T2EP20122-0011).

## Impact Statement

This paper presents work whose goal is to advance the field of Machine Learning. There are many potential societal consequences of our work, none of which we feel must be specifically highlighted here.

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

# Appendix

# A. Satori's Demo Examples

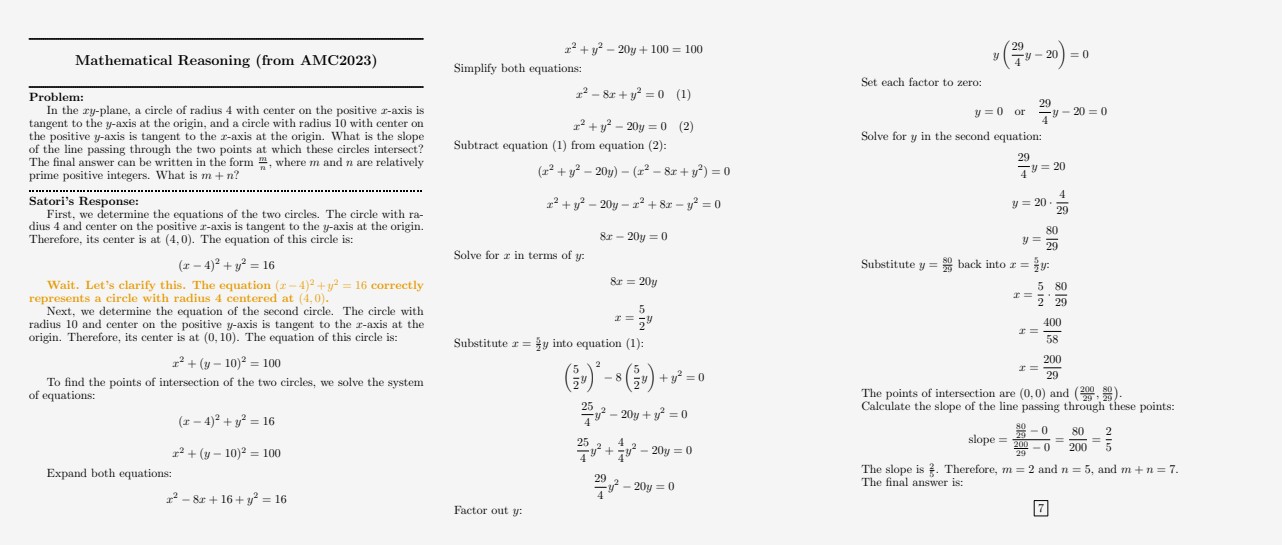

*Figure 6:* **Math Domain Example.** Satori verifies the correctness of the intermediate steps and proceeds to the next reasoning step.

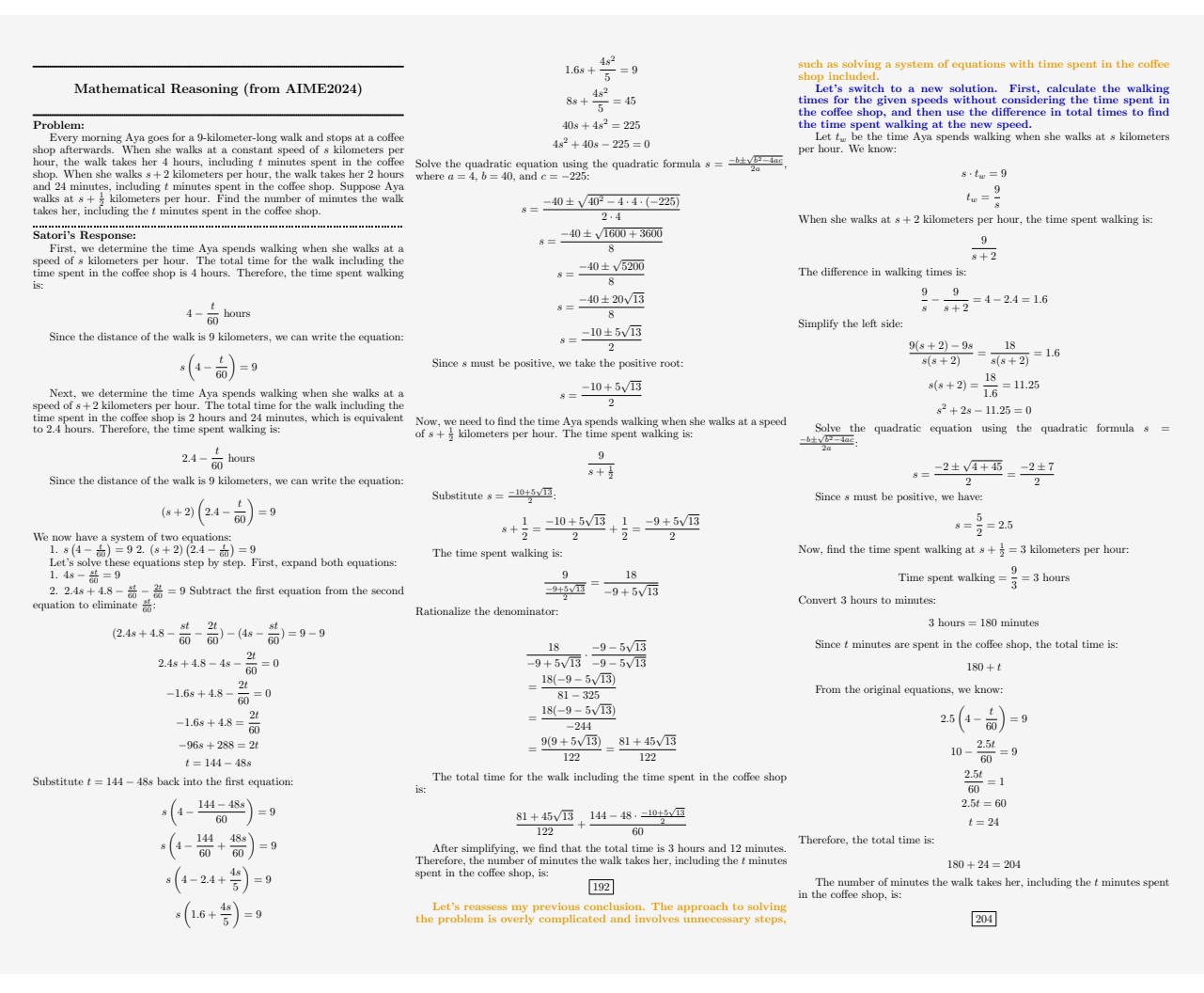

*Figure 7:* **Math Domain Example.** Satori identifies the mistakes in the previous solution and proposes an alternative correct solution.

---

**Mathematical Reasoning (from MATH)**

---

**Problem:**
In the diagram, the four points have coordinates $A(0,1)$, $B(1,3)$, $C(5,2)$, and $D(4,0)$. What is the area of quadrilateral $ABCD$?

```
[asy]
// draw axis
real low = -1; real high = 6;
draw((low, 0)--(high, 0), Arrow); draw((0, low)--(0, high - 1.5), Arrow);
label("$x$", (high, 0), E); label("$y$", (0, high - 1.5), N);

// draw quadrilateral
pair a = (0, 1); pair b = (1, 3); pair c = (5, 2); pair d = (4, 0);
draw(a--b--c--d--cycle);
label("$A$", a, W); label("$B$", b, N); label("$C$", c, E); label("$D$", d, S);
[/asy]
```

**Satori's Response:**
To find the area of quadrilateral $ABCD$ with vertices $A(0,1)$, $B(1,3)$, $C(5,2)$, and $D(4,0)$, we can use the Shoelace Theorem (also known as Gauss's area formula for polygons). The Shoelace Theorem states that for a polygon with vertices $(x_1, y_1), (x_2, y_2), \ldots, (x_n, y_n)$, the area $A$ is given by:

$$A = \frac{1}{2} \left| \sum_{i=1}^{n-1} (x_i y_{i+1} - y_i x_{i+1}) + (x_n y_1 - y_n x_1) \right|$$

First, we list the coordinates of the vertices in order: $A(0,1)$, $B(1,3)$, $C(5,2)$, and $D(4,0)$.
We apply the Shoelace Theorem:

$$A = \frac{1}{2} |(0 \cdot 3 + 1 \cdot 2 + 5 \cdot 0 + 4 \cdot 1) - (1 \cdot 1 + 3 \cdot 5 + 2 \cdot 4 + 0 \cdot 0)|$$

Calculate the sum of the products of the coordinates:

$$\text{Sum of } x_i y_{i+1} = 0 \cdot 3 + 1 \cdot 2 + 5 \cdot 0 + 4 \cdot 1 = 0 + 2 + 0 + 4 = 6$$

$$\text{Sum of } y_i x_{i+1} = 1 \cdot 1 + 3 \cdot 5 + 2 \cdot 4 + 0 \cdot 0 = 1 + 15 + 8 + 0 = 24$$

Substitute these sums into the Shoelace Theorem formula:

$$A = \frac{1}{2} |6 - 24| = \frac{1}{2} |-18| = \frac{1}{2} \times 18 = 9$$

Therefore, the area of quadrilateral $ABCD$ is:

$$\boxed{9}$$

**Wait. Let's clarify this. The application of the Shoelace Theorem is correct, and the calculations for the sums of the products of the coordinates are accurate, leading to the correct area of 9.**
Let's verify the calculation using another method, such as dividing the quadrilateral into two triangles and finding the area of each triangle.
First, we find the area of triangle $ABD$:
- vertices: $A(0,1)$, $B(1,3)$, $D(4,0)$
Using the formula for the area of a triangle with vertices $(x_1, y_1)$, $(x_2, y_2)$, and $(x_3, y_3)$:

$$\text{Area} = \frac{1}{2} |x_1(y_2 - y_3) + x_2(y_3 - y_1) + x_3(y_1 - y_2)|$$

For triangle $ABD$:

$$\text{Area}_{ABD} = \frac{1}{2} |0(3-0) + 1(0-1) + 4(1-3)| = \frac{1}{2} |0 - 1 - 8| = \frac{1}{2} |-9| = \frac{9}{2}$$

Next, we find the area of triangle $BCD$:
- vertices: $B(1,3)$, $C(5,2)$, $D(4,0)$
Using the same formula:

$$\text{Area}_{BCD} = \frac{1}{2} |1(2-0) + 5(0-3) + 4(3-2)| = \frac{1}{2} |2 - 15 + 4| = \frac{1}{2} |-9| = \frac{9}{2}$$

Finally, we add the areas of the two triangles to get the area of quadrilateral $ABCD$:

$$\text{Area}_{ABCD} = \text{Area}_{ABD} + \text{Area}_{BCD} = \frac{9}{2} + \frac{9}{2} = 9$$

Thus, the area of quadrilateral $ABCD$ is indeed:

$$\boxed{9}$$

*Figure 8:* **Math Domain Example.** Satori verifies the correctness of previous solution and initiates a different solution.

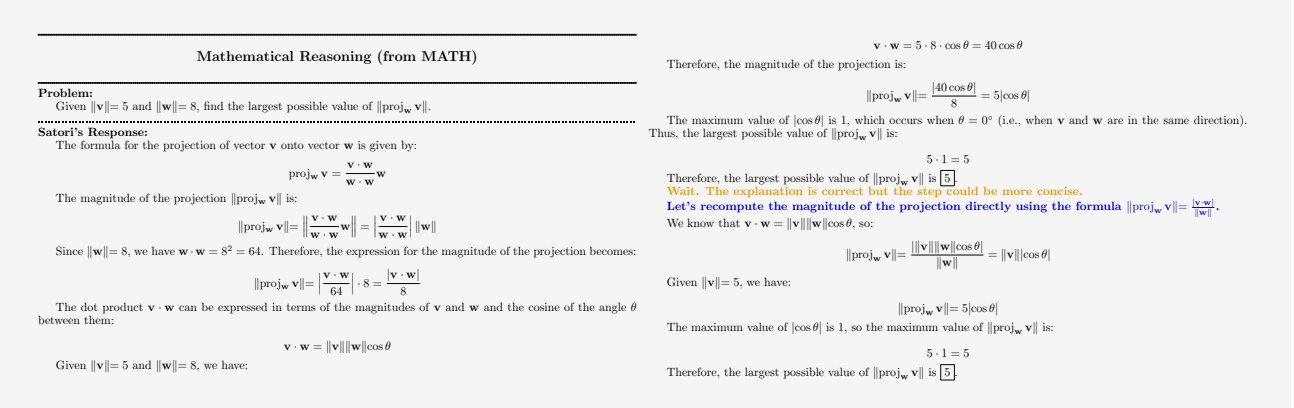

---

**Mathematical Reasoning (from MATH)**

---

**Problem:**
Given $\|\mathbf{v}\| = 5$ and $\|\mathbf{w}\| = 8$, find the largest possible value of $\|\text{proj}_{\mathbf{w}} \mathbf{v}\|$.

**Satori's Response:**
The formula for the projection of vector $\mathbf{v}$ onto vector $\mathbf{w}$ is given by:

$$\text{proj}_{\mathbf{w}} \mathbf{v} = \frac{\mathbf{v} \cdot \mathbf{w}}{\mathbf{w} \cdot \mathbf{w}} \mathbf{w}$$

The magnitude of the projection $\|\text{proj}_{\mathbf{w}} \mathbf{v}\|$ is:

$$\|\text{proj}_{\mathbf{w}} \mathbf{v}\| = \left\| \frac{\mathbf{v} \cdot \mathbf{w}}{\mathbf{w} \cdot \mathbf{w}} \mathbf{w} \right\| = \left| \frac{\mathbf{v} \cdot \mathbf{w}}{\mathbf{w} \cdot \mathbf{w}} \right| \|\mathbf{w}\|$$

Since $\|\mathbf{w}\| = 8$, we have $\mathbf{w} \cdot \mathbf{w} = 8^2 = 64$. Therefore, the expression for the magnitude of the projection becomes:

$$\|\text{proj}_{\mathbf{w}} \mathbf{v}\| = \left| \frac{\mathbf{v} \cdot \mathbf{w}}{64} \right| \cdot 8 = \frac{|\mathbf{v} \cdot \mathbf{w}|}{8}$$

The dot product $\mathbf{v} \cdot \mathbf{w}$ can be expressed in terms of the magnitudes of $\mathbf{v}$ and $\mathbf{w}$ and the cosine of the angle $\theta$ between them:

$$\mathbf{v} \cdot \mathbf{w} = \|\mathbf{v}\| \|\mathbf{w}\| \cos \theta$$

Given $\|\mathbf{v}\| = 5$ and $\|\mathbf{w}\| = 8$, we have:

$$\mathbf{v} \cdot \mathbf{w} = 5 \cdot 8 \cdot \cos \theta = 40 \cos \theta$$

Therefore, the magnitude of the projection is:

$$\|\text{proj}_{\mathbf{w}} \mathbf{v}\| = \frac{|40 \cos \theta|}{8} = 5|\cos \theta|$$

The maximum value of $|\cos \theta|$ is 1, which occurs when $\theta = 0°$ (i.e., when $\mathbf{v}$ and $\mathbf{w}$ are in the same direction). Thus, the largest possible value of $\|\text{proj}_{\mathbf{w}} \mathbf{v}\|$ is:

$$5 \cdot 1 = 5$$

Therefore, the largest possible value of $\|\text{proj}_{\mathbf{w}} \mathbf{v}\|$ is $\boxed{5}$.
**Wait. The explanation is correct but the step could be more concise.**
**Let's recompute the magnitude of the projection directly using the formula** $\|\text{proj}_{\mathbf{w}} \mathbf{v}\| = \frac{|\mathbf{v} \cdot \mathbf{w}|}{\|\mathbf{w}\|}$.
We know that $\mathbf{v} \cdot \mathbf{w} = \|\mathbf{v}\| \|\mathbf{w}\| \cos \theta$, so:

$$\|\text{proj}_{\mathbf{w}} \mathbf{v}\| = \frac{|\|\mathbf{v}\| \|\mathbf{w}\| \cos \theta|}{\|\mathbf{w}\|} = \|\mathbf{v}\| |\cos \theta|$$

Given $\|\mathbf{v}\| = 5$, we have:

$$\|\text{proj}_{\mathbf{w}} \mathbf{v}\| = 5|\cos \theta|$$

The maximum value of $|\cos \theta|$ is 1, so the maximum value of $\|\text{proj}_{\mathbf{w}} \mathbf{v}\|$ is:

$$5 \cdot 1 = 5$$

Therefore, the largest possible value of $\|\text{proj}_{\mathbf{w}} \mathbf{v}\|$ is $\boxed{5}$.

*Figure 9:* **Math Domain Example.** Satori verifies the correctness of previous solution and further explores a simpler solution.

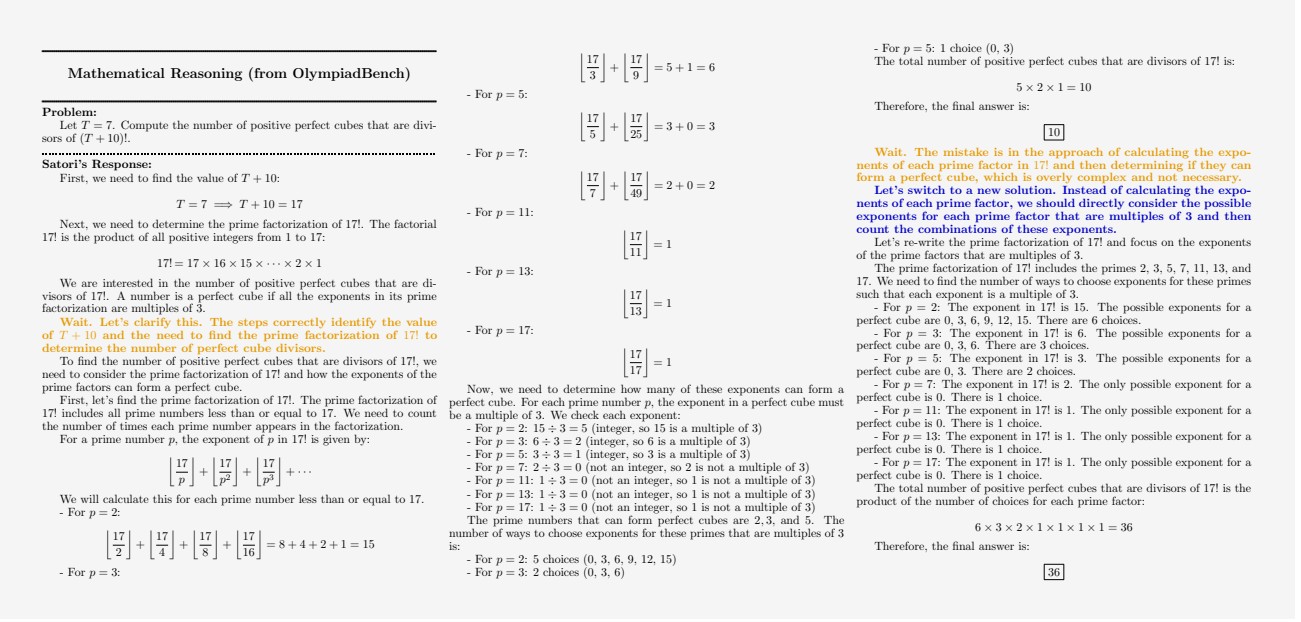

*Figure 10:* **Math Domain Example.** 1) Satori verifies the correctness of intermediate steps in early stage. 2) Satori realizes that the pervious solution is actually erroneous and then proposes an alternative correct solution.

---

**Logical Reasoning (from BoardgameQA)**

---

**Problem:**
Solve the following problem. Your final answer should be "True", "False", or "Uncertain".

A few players are playing a boardgame. The current state of the game is as follows. The liger has 1 friend that is easy going and 3 friends that are not. The liger is a physiotherapist. The seal has a football with a radius of 22 inches, and is a teacher assistant. The seal hides the cards that she has from the bee but does not build a power plant near the green fields of the husky.

And the rules of the game are as follows. Rule1: If the liger works in healthcare, then the liger does not manage to convince the pelikan. Rule2: The seal will bring an oil tank for the pelikan if it (the seal) has a football that fits in a 52.3 x 43.6 x 36.1 inches box. Rule3: Regarding the seal, if it works in education, then we can conclude that it brings an oil tank for the pelikan. Rule4: If you see that something hides the cards that she has from the bee but does not build a power plant near the green fields of the husky, what can you certainly conclude? You can conclude that it does not bring an oil tank for the pelikan. Rule5: For the pelikan, if you have two pieces of evidence 1) the liger manages to convince the pelikan and 2) the seal brings an oil tank for the pelikan, then you can add "pelikan will never bring an oil tank for the dragon" to your conclusions. Rule6: The liger will manage to persuade the pelikan if it (the liger) has fewer than five friends.

Rule2 is preferred over Rule4. Rule3 is preferred over Rule4. Rule6 is preferred over Rule1.

Based on the game state and the rules and preferences, does the pelikan bring an oil tank for the dragon?

...............................................................................................................................................................................

**Satori's Response:**
First, we analyze the given information and rules:
1. The liger has 1 friend that is easy going and 3 friends that are not.
2. The liger is a physiotherapist.
3. The seal has a football with a radius of 22 inches.
4. The seal hides the cards that she has from the bee but does not build a power plant near the green fields of the husky.
5. Rule1: If the liger works in healthcare, then the liger does not manage to convince the pelikan.
6. Rule2: The seal will bring an oil tank for the pelikan if it (the seal) has a football that fits in a 52.3 x 43.6 x 36.1 inches box.
7. Rule3: Regarding the seal, if it works in education, then we can conclude that it brings an oil tank for the pelikan.
8. Rule4: If you see that something hides the cards that she has from the bee but does not build a power plant near the green fields of the husky, what can you certainly conclude? You can conclude that it does not bring an oil tank for the pelikan.
9. Rule5: For the pelikan, if you have two pieces of evidence 1) the liger manages to convince the pelikan and 2) the seal brings an oil tank for the pelikan, then you can add "pelikan will never bring an oil tank for the dragon" to your conclusions.
10. Rule6: The liger will manage to persuade the pelikan if it (the liger) has fewer than five friends.
*Wait. The mistake is that the solution starts analyzing the given information and rules without directly addressing the question of whether the pelikan brings an oil tank for the dragon.*
*Let's re-evaluate the problem step-by-step to determine if the pelikan brings an oil tank for the dragon.*
1. **Liger's Friends and Occupation:**
- The liger has 1 friend that is easy going and 3 friends that are not.
- The liger is a physiotherapist.
- According to Rule6: The liger will manage to persuade the pelikan if it (the liger) has fewer than five friends.
- Since the liger has 4 friends (1 easy going + 3 not), the liger will manage to convince the pelikan.
2. **Seal's Football and Box:**
- The seal has a football with a radius of 22 inches.
- The dimensions of the box are 52.3 x 43.6 x 36.1 inches.
- The volume of the football is given by $V = \frac{4}{3}\pi r^3$.

$$V = \frac{4}{3}\pi(22)^3 \approx \frac{4}{3}\pi(10648) \approx 44922.67 \text{ cubic inches}$$

- The volume of the box is:

$$V_{\text{box}} = 52.3 \times 43.6 \times 36.1 \approx 83257.69 \text{ cubic inches}$$

- Since the volume of the football is less than the volume of the box, the seal's football fits in the box.
- According to Rule2: The seal will bring an oil tank for the pelikan if it (the seal) has a football that fits in a 52.3 x 43.6 x 36.1 inches box.
- Therefore, the seal will bring an oil tank for the pelikan.
3. **Conclusion Based on Rules:**
- From Rule4: If you see that something hides the cards that she has from the bee but does not build a power plant near the green fields of the husky, what can you certainly conclude? You can conclude that it does not bring an oil tank for the pelikan.
- The seal hides the cards that she has from the bee but does not build a power plant near the green fields of the husky.
- Therefore, the seal does not bring an oil tank for the pelikan.
- This creates a contradiction with Rule2, which states that the seal will bring an oil tank for the pelikan.
- Given the rules and the information, the most logical conclusion is that the seal will bring an oil tank for the pelikan, despite the contradiction.
4. **Final Conclusion:**
- Based on the analysis of the rules and the given information, the seal will bring an oil tank for the pelikan.
Therefore, the final answer is: [True].
*Wait. The mistake is that the solution starts analyzing the given information and rules without directly addressing the specific question about the pelikan bringing an oil tank for the dragon.*
*Let's simplify the approach and focus on the direct question. We need to determine if the pelikan brings an oil tank for the dragon based on the given rules and the information about the liger and the seal.*
Based on the given information:
- The liger has 4 friends (1 easy going and 3 not), so the liger will manage to convince the pelikan (Rule6).
- The seal has a football with a radius of 22 inches, which fits in a 52.3 x 43.6 x 36.1 inches box, so the seal will bring an oil tank for the pelikan (Rule2).
- The seal hides the cards that she has from the bee but does not build a power plant near the green fields of the husky, so according to Rule4, the seal does not bring an oil tank for the pelikan.
There is a contradiction between Rule2 and Rule4. However, since Rule4 states a condition that leads to a specific conclusion if both conditions are met, we should consider the direct implications of each rule.
Given the contradiction, we need to re-evaluate the logical flow. The key is to recognize that Rule4 provides a clear condition that negates the oil tank bringing behavior if both conditions are met. Since the seal meets the condition of hiding cards and not building a power plant, Rule4 takes precedence over Rule2.
Therefore, the pelikan does not bring an oil tank for the dragon.
The final answer is: [False].

*Figure 11:* **Out-of-domain Example.** 1) Satori identifies the potential mistakes in intermediate steps and initiates another solution. 2) Satori realizes that the pervious solution is still erroneous and then proposes an alternative correct solution.

---

**Commonsense Reasoning (from StrategyQA)**

---

**Problem:**
You are given the following facts:
1. Christopher Reeve suffered a serious spinal cord injury that left him a quadriplegic.
2. Christopher Reeve required a portable ventilator after his injury.
3. At one point, Christopher Reeve's treatment was costing $400,000 yearly.
Based on the facts above, answer the following question. Your final answer should be either "True" or "False".
If he were poor, would Christopher Reeve have lived?

...............................................................................................................................................................................

**Satori's Response:**
The question asks whether Christopher Reeve would have lived if he were poor.
To answer this, we need to consider the impact of his spinal cord injury and the cost of his treatment.
Christopher Reeve's spinal cord injury left him a quadriplegic, which means he was paralyzed from the neck down. This made it difficult for him to move and perform many daily activities.
The fact that he required a portable ventilator indicates that he was dependent on mechanical assistance for breathing.
The cost of his treatment was $400,000 yearly, which is a significant amount of money.
If Christopher Reeve were poor, he would likely not have been able to afford this level of medical care.
Without the necessary medical treatment, it is highly unlikely that he would have been able to live a normal life.
*Wait. The question is not about the impact of his condition or the cost of treatment on his ability to live, but rather if he would have lived given the circumstances.*
*Let's switch to a new solution. Focus on the fact that the question asks if he would have lived if he were poor, and consider whether the cost of treatment alone would have been a determining factor in his survival.*
Let's re-evaluate the question more carefully. The question asks if Christopher Reeve would have lived if he were poor.
The key factors to consider are:
1. The severity of his spinal cord injury.
2. The cost of his treatment.
3. The impact of poverty on his ability to access and afford medical care.
Christopher Reeve's spinal cord injury was severe, and he required specialized medical equipment and ongoing treatment. The cost of $400,000 yearly was substantial and likely beyond the means of many people, especially those who are poor.
If Christopher Reeve were poor, he would likely not have been able to afford this level of medical care. The cost of his treatment would have been a significant financial burden.
Given the severity of his condition and the high cost of treatment, it is reasonable to conclude that he would not have been able to live a normal life if he were poor.
Therefore, the final answer is: [False].

*Figure 12:* **Out-of-domain Example.** Satori identifies the potential mistakes in intermediate steps and initiates another correct solution.

---

**Domain-specific Reasoning (from MMLUPro - Chemistry)**

---

**Problem:**

Assume all gases are perfect unless stated otherwise. Unless otherwise stated, thermodynamic data are for 298.15 K. A strip of magnesium of mass 15 g is placed in a beaker of dilute hydrochloric acid. Calculate the work done by the system as a result of the reaction. The atmospheric pressure is 1.0 atm and the temperature 25°C.
(A) −1.5kJ
(B) 0kJ (No work done)
(C) −3.5kJ
(D) −0.5kJ
(E) −2.0kJ
(F) 2.5kJ (Positive work done by the system)
(G) −3.0kJ
(H) −1.0kJ
(I) −2.5kJ
(J) 1.0kJ (Positive work done by the system)

**Satori's Response:**

The reaction between magnesium and hydrochloric acid is:

$$Mg + 2HCl \rightarrow MgCl_2 + H_2$$

The work done by the system as a result of the reaction is given by the formula:

$$w = -P\Delta V$$

where $P$ is the atmospheric pressure and $\Delta V$ is the change in volume of the system.
First, we need to determine the amount of hydrogen gas produced. The balanced equation shows that 1 mole of magnesium produces 1 mole of hydrogen gas.
The molar mass of magnesium is 24.31 g/mol. Therefore, the number of moles of magnesium in 15 g is:

$$\text{moles of Mg} = \frac{15 \text{ g}}{24.31 \text{ g/mol}} \approx 0.617 \text{ mol}$$

Since 1 mole of magnesium produces 1 mole of hydrogen gas, the number of moles of hydrogen gas produced is also 0.617 mol.
The volume of hydrogen gas produced at 25°C (298.15 K) and 1.0 atm can be calculated using the ideal gas law:

$$V = nRT/P$$

where $n$ is the number of moles of gas, $R$ is the ideal gas constant (0.08206 L·atm/(mol·K)), $T$ is the temperature in Kelvin, and $P$ is the pressure in atmospheres.
**Wait. Let's clarify this. The steps correctly identify the reaction, the formula for work, and the calculation of the moles of magnesium and hydrogen gas produced, but the volume calculation is unnecessary for determining the work done by the system in this context.**
The work done by the system is zero because the reaction occurs in a beaker and the volume change is negligible under the given conditions. The system does not expand or compress against the external pressure.
Therefore, the work done by the system is: $w = 0$ kJ
The correct answer is: $\boxed{B}$
**Wait. The mistake is that the volume change is incorrectly assumed to be negligible, leading to the incorrect conclusion that no work is done.**
**Let's switch to a new solution. Calculate the volume of hydrogen gas produced using the ideal gas law and then use it to find the work done by the system.**
Let's re-evaluate the problem step-by-step.
1. **Determine the amount of hydrogen gas produced:**
The balanced equation is:

$$Mg + 2HCl \rightarrow MgCl_2 + H_2$$

From the equation, 1 mole of Mg produces 1 mole of $H_2$.
The molar mass of Mg is 24.31 g/mol. Therefore, the number of moles of Mg in 15 g is:

$$\text{moles of Mg} = \frac{15 \text{ g}}{24.31 \text{ g/mol}} \approx 0.617 \text{ mol}$$

So, the number of moles of $H_2$ produced is also 0.617 mol.
2. **Calculate the volume of hydrogen gas produced:**
Using the ideal gas law $V = nRT/P$, where $n = 0.617$ mol, $R = 0.08206$ L·atm/(mol·K), $T = 298.15$ K, and $P = 1.0$ atm:

$$V = 0.617 \text{ mol} \times 0.08206 \text{ L·atm/(mol·K)} \times 298.15 \text{ K}/1.0 \text{ atm} \approx 15.2 \text{ L}$$

3. **Calculate the work done by the system:**
The work done by the system is given by $w = -P\Delta V$. Since the reaction occurs in a beaker and the volume change is significant, we need to consider the external pressure and the change in volume.

$$w = -1.0 \text{ atm} \times 15.2 \text{ L} = -15.2 \text{ L·atm}$$

Convert L·atm to kJ using the conversion factor 1 L·atm = 101.325 J:

$$w = -15.2 \text{ L·atm} \times \frac{101.325 \text{ J}}{1 \text{ L·atm}} \times \frac{1 \text{ kJ}}{1000 \text{ J}} \approx -1.54 \text{ kJ}$$

Therefore, the work done by the system is approximately −1.5 kJ.
The correct answer is: $\boxed{A}$

*Figure 13:* **Out-of-domain Example.** 1) Satori verifies the correctness of intermediate steps in early stage. 2) Satori realizes that the pervious solution is actually erroneous and then proposes an alternative correct solution.

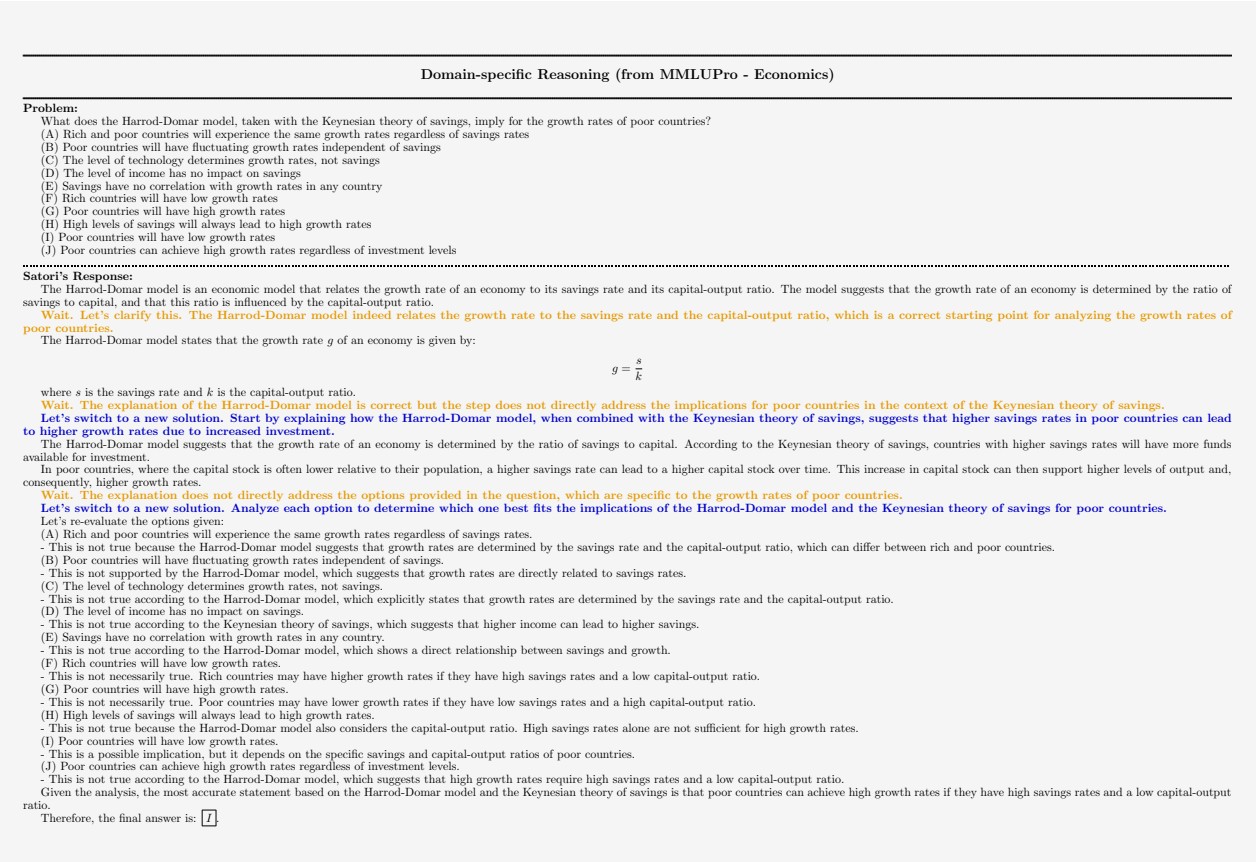

**Domain-specific Reasoning (from MMLUPro - Economics)**

**Problem:**
What does the Harrod-Domar model, taken with the Keynesian theory of savings, imply for the growth rates of poor countries?
(A) Rich and poor countries will experience the same growth rates regardless of savings rates
(B) Poor countries will have fluctuating growth rates independent of savings
(C) The level of technology determines growth rates, not savings
(D) The level of income has no impact on savings
(E) Savings have no correlation with growth rates in any country
(F) Rich countries will have low growth rates
(G) Poor countries will have high growth rates
(H) High levels of savings will always lead to high growth rates
(I) Poor countries will have low growth rates
(J) Poor countries can achieve high growth rates regardless of investment levels

**Satori's Response:**
The Harrod-Domar model is an economic model that relates the growth rate of an economy to its savings rate and its capital-output ratio. The model suggests that the growth rate of an economy is determined by the ratio of savings to capital, and that this ratio is influenced by the capital-output ratio.
Wait. Let's clarify this. The Harrod-Domar model indeed relates the growth rate to the savings rate and the capital-output ratio, which is a correct starting point for analyzing the growth rates of poor countries.
The Harrod-Domar model states that the growth rate $g$ of an economy is given by:

$$g = \frac{s}{k}$$

where $s$ is the savings rate and $k$ is the capital-output ratio.
Wait. The explanation of the Harrod-Domar model is correct but the step does not directly address the implications for poor countries in the context of the Keynesian theory of savings.
Let's switch to a new solution. Start by explaining how the Harrod-Domar model, when combined with the Keynesian theory of savings, suggests that higher savings rates in poor countries can lead to higher growth rates due to increased investment.
The Harrod-Domar model suggests that the growth rate of an economy is determined by the ratio of savings to capital. According to the Keynesian theory of savings, countries with higher savings rates will have more funds available for investment.
In poor countries, where the capital stock is often lower relative to their population, a higher savings rate can lead to a higher capital stock over time. This increase in capital stock can then support higher levels of output and, consequently, higher growth rates.
Wait. The explanation does not directly address the options provided in the question, which are specific to the growth rates of poor countries.
Let's switch to a new solution. Analyze each option to determine which one best fits the implications of the Harrod-Domar model and the Keynesian theory of savings for poor countries.
Let's re-evaluate the options given:
(A) Rich and poor countries will experience the same growth rates regardless of savings rates.
- This is not true because the Harrod-Domar model suggests that growth rates are determined by the savings rate and the capital-output ratio, which can differ between rich and poor countries.
(B) Poor countries will have fluctuating growth rates independent of savings.
- This is not supported by the Harrod-Domar model, which suggests that growth rates are directly related to savings rates.
(C) The level of technology determines growth rates, not savings.
- This is not true according to the Harrod-Domar model, which explicitly states that growth rates are determined by the savings rate and the capital-output ratio.
(D) The level of income has no impact on savings.
- This is not true according to the Keynesian theory of savings, which suggests that higher income can lead to higher savings.
(E) Savings have no correlation with growth rates in any country.
- This is not true according to the Harrod-Domar model, which shows a direct relationship between savings and growth.
(F) Rich countries will have low growth rates.
- This is not necessarily true. Rich countries may have higher growth rates if they have high savings rates and a low capital-output ratio.
(G) Poor countries will have high growth rates.
- This is not necessarily true. Poor countries may have lower growth rates if they have low savings rates and a high capital-output ratio.
(H) High levels of savings will always lead to high growth rates.
- This is not true because the Harrod-Domar model also considers the capital-output ratio. High savings rates alone are not sufficient for high growth rates.
(I) Poor countries will have low growth rates.
- This is a possible implication, but it depends on the specific savings and capital-output ratios of poor countries.
(J) Poor countries can achieve high growth rates regardless of investment levels.
- This is not true according to the Harrod-Domar model, which suggests that high growth rates require high savings rates and a low capital-output ratio.
Given the analysis, the most accurate statement based on the Harrod-Domar model and the Keynesian theory of savings is that poor countries can achieve high growth rates if they have high savings rates and a low capital-output ratio.
Therefore, the final answer is: $\boxed{I}$.

*Figure 14:* **Out-of-domain Example.** Satori engages in multiple self-reflection processes during intermediate reasoning steps.

---

**Code Reasoning (from CRUXEval)**

**Problem:**
You are given the following function:

```python
def f(text, sep, maxsplit):
    splitted = text.rsplit(sep, maxsplit)
    length = len(splitted)
    new_splitted = splitted[:length // 2]
    new_splitted.reverse()
    new_splitted += splitted[length // 2:]
    return sep.join(new_splitted)
```

You are also given the following input (the order of the input is the same as the order of the input variables in the function):

'ertubwi', 'p', 5

With the given function and input, what would be the output?

**Satori's Response:**
First, we need to understand the function $f$ and how it processes the given input. The function $f$ takes three arguments: a string `text`, a separator `sep`, and a maximum number of splits `maxsplit`. It splits the string `text` into parts using the separator `sep` up to `maxsplit` times, reverses the first half of the resulting list of parts, and then concatenates the reversed first half with the second half using the separator `sep`.
Wait. Let's clarify this. The function correctly processes the input by splitting the string, reversing the first half, and concatenating the parts, which aligns with the function's intended behavior.
Let's break down the function step by step with the given input: 'ertubwi', 'p', 5.
1. **Split the string:** The function uses `text.rsplit(sep, maxsplit)`. Here, `text = 'ertubwi'`, `sep = 'p'`, and `maxsplit = 5`. Since 'ertubwi' does not contain the separator 'p', `text.rsplit(sep, maxsplit)` will split the string into a list containing the entire string and four empty strings (because `maxsplit` is 5, but there are no occurrences of 'p'):
   `splitted = ['ertubwi', '', '', '', '']`
2. **Determine the length:** The length of `splitted` is 5.
3. **Split the list into two halves:** Since the length is 5, the first half will be the first 2 elements (index 0 and 1), and the second half will be the remaining elements (index 2, 3, and 4):
   `new_splitted = splitted[:5 // 2] = ['ertubwi', '']`
   `new_splitted += splitted[5 // 2:] = ['ertubwi', '', '', '', '']`
4. **Reverse the first half:** Reverse the elements in `new_splitted[:2]`:
   `new_splitted[:2] = ['', 'ertubwi']`
5. **Concatenate the parts:** Now, concatenate the reversed first half with the second half using the separator 'p':
   `new_splitted = ['', 'ertubwi', '', '', '']`
   `result = 'p'.join(new_splitted) = 'p' + '' + 'ertubwi' + 'p' + '' + 'p' + '' + 'p' + '' = 'p ertubwi p p p p'`
However, since the second half should only include the remaining elements after the first half, we need to re-evaluate the correct approach. Let's consider the correct approach:
1. **Split the string:** Since 'ertubwi' does not contain 'p', we need to consider the correct approach to split and reverse.
Given the function's behavior and the input, the correct approach is to split the string into two equal parts and reverse the first part.
1. **Split the string:** 'ertubwi' into 'ertu' and 'bwi'.
2. **Reverse the first part:** 'ertu' reversed is 'uert'.
3. **Concatenate: 'uert' and 'bwi' with separator 'p':** 'uertpbwi'.
Therefore, the final answer is: `uertpbwi`.
Wait. The mistake is in the incorrect handling of the split and the subsequent concatenation, leading to an incorrect final string.
Let's switch to a new solution. We should correctly split the string 'ertubwi' into parts using the separator 'p', which in this case does not occur, so we should consider the string as a whole and apply the function's logic directly.
Let's re-evaluate the function step-by-step with the given input: 'ertubwi', 'p', 5.
1. **Split the string:** Since 'ertubwi' does not contain the separator 'p', the function will split it into a list with one element: `['ertubwi']`.
2. **Determine the length:** The length of the list is 1.
3. **Split the list into two halves:** Since the length is odd, the first half will have 0 elements, and the second half will have 1 element.
4. **Reverse the first half:** The first half is empty, so it remains empty.
5. **Concatenate the parts:** Concatenate the reversed first half with the second half using the separator 'p': `'' + 'ertubwi' = 'ertubwi'`.
Therefore, the final answer is: `ertubwi`.

*Figure 15:* **Out-of-domain Example.** 1) Satori verifies the correctness of intermediate steps in early stage. 2) Satori realizes that the pervious solution is actually erroneous and then proposes an alternative correct solution.

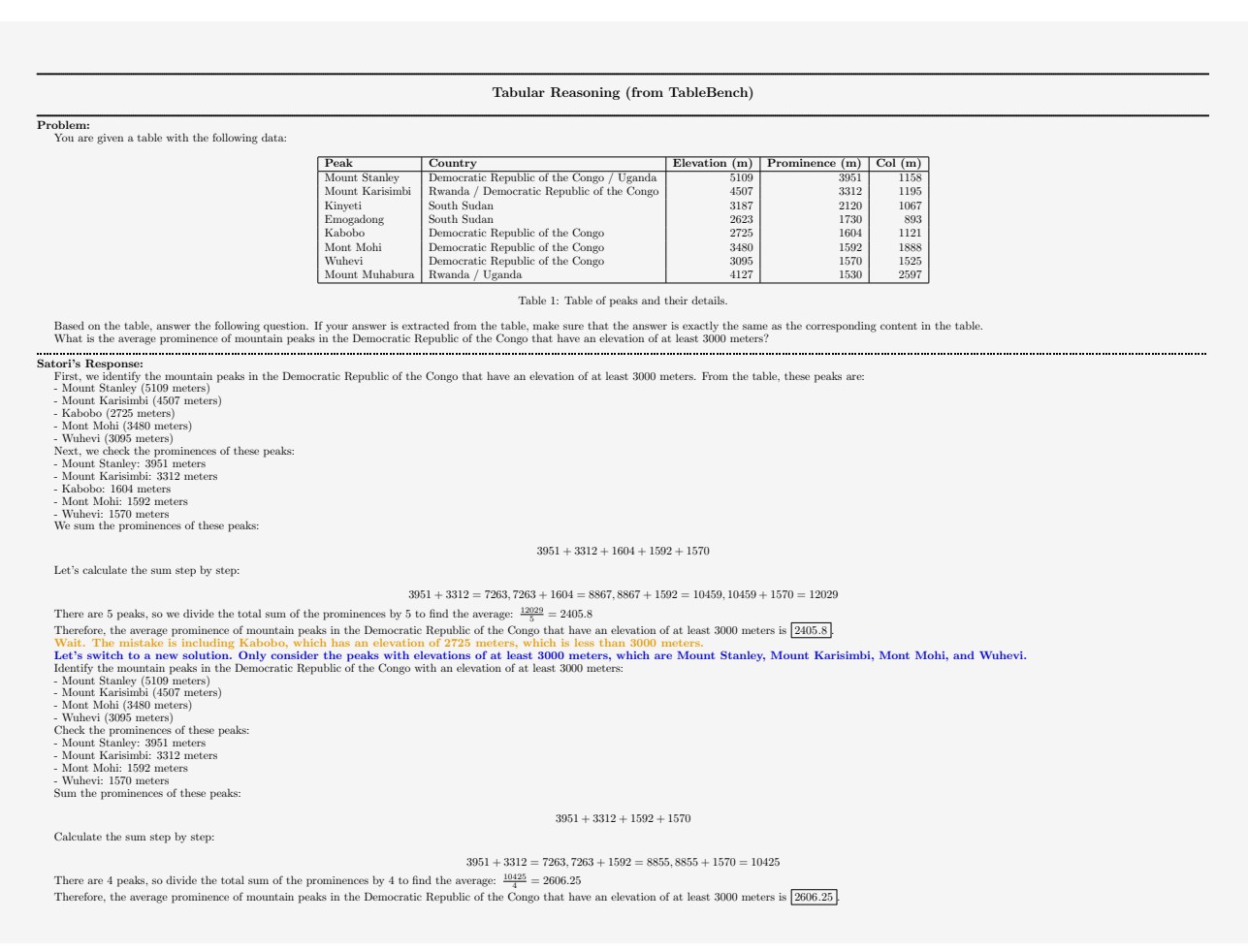

*Figure 16:* **Out-of-domain Example.** Satori identifies the mistakes in previous solution and proposes an alternative correct solution.

# B. Additional Related Work

## B.1. Post-training LLMs for Reasoning

State-of-the-art LLMs have achieved human-level and, in some cases, "superhuman" performance across diverse reasoning benchmarks. These benchmarks span various domains, including mathematics (Cobbe et al., 2021; Hendrycks et al., 2021a; Glazer et al., 2024; Mirzadeh et al., 2024), programming (Chen et al., 2021; Austin et al., 2021; Zhuo et al., 2024; Dai et al., 2024; Jimenez et al., 2024; Gu et al., 2024), logical reasoning (Han et al., 2024; Liu et al., 2020; 2023; Tafjord et al., 2021), commonsense reasoning (Geva et al., 2021; Talmor et al., 2018), algorithmic reasoning (Veličković et al., 2022; Markeeva et al., 2024; Qi et al., 2024b; Fan et al., 2024), semi-structured data (Wu et al., 2024a; Lu et al., 2022), scientific knowledge (Wang et al., 2024c; Rein et al., 2024), and world knowledge (Hendrycks et al., 2021b; Wang et al., 2024b; Srivastava et al., 2023; Liang et al., 2023; Phan et al., 2025).

Recent advancements have concentrated on extensive post-training to enhance LLMs' reasoning abilities. One research direction in this area involves constructing instruction-tuning datasets annotated with high-quality CoT-like reasoning chains. These datasets are created either through extensive human annotation (Hendrycks et al., 2021a; Saunders et al., 2022b; Yue et al., 2024) or by distilling data from more advanced models (Yu et al., 2024; Toshniwal et al., 2024b;a; Ding et al., 2024; Luo et al., 2023; Huang et al., 2024a; Zhao et al., 2024; Abdin et al., 2024; Min et al., 2024). However, human annotation is resource-intensive, and model-generated data inherently caps the student model's potential at the level of the teacher model.

More recent research has focused on self-improvement approaches, where models are trained on data generated by themselves (Zelikman et al., 2022; 2024; Singh et al., 2024; Zhang et al., 2024a). While self-training mitigates the reliance on external resources, it has raised concerns about potential "model collapse", a phenomenon where the iterative use of model-generated data degrades performance (Shumailov et al., 2023; Wu et al., 2024b). Additionally, reinforcement learning methods, particularly those based on Proximal Policy Optimization (PPO) (Schulman et al., 2017a; Ouyang et al., 2022), have been explored to enhance reasoning capabilities. These approaches typically utilize reward models, such as Process-Reward Models (PRMs) or Outcome-Reward Models (ORMs), to guide the learning process (Sun et al., 2024; Wang et al., 2024a; Yuan et al., 2024), resulting in significant performance improvements compared to supervised fine-tuning.

## B.2. Enabling LLMs with Searching Abilities

Chain-of-Thought (CoT) prompting (Wei et al., 2022) demonstrated its potential to improve reasoning but lacked mechanisms to correct previous errors once committed. To address this, subsequent work proposed more sophisticated methods (Yao et al., 2024; Shinn et al., 2024; Besta et al., 2024; Madaan et al., 2024; Yang et al., 2024b) that prompt LLMs to search for solutions via forward exploration, backtracking from errors, and finding alternate paths. Heuristic search methods (Hao et al., 2023; Qi et al., 2024a) have also been adopted to enable more effective exploration of high-quality solutions. However, prompting-based approaches improve task-specific performance without fundamentally enhancing the LLM's intrinsic reasoning capabilities. Moreover, recent work has pointed out the inherent difficulties of current LLMs in conducting effective self-correction (Huang et al., 2024b; Zhang et al., 2024b; Kamoi et al., 2024).

Recent research has pivoted toward training LLMs explicitly for exploration and backtracking. A large body of work has focused on enabling *trajectory-level search* abilities, which train LLMs to iteratively identify errors in their previous complete responses and produce improved responses, relying on either human-annotated revisions (Saunders et al., 2022a) or model-generated data (Kumar et al., 2024; Qu et al., 2024; Havrilla et al., 2024) as training data. Another line of research has investigated *step-level search* techniques, which induce more fine-grained and real-time correction that enables LLMs to identify and correct mistakes once they occur. Some achieve this by leveraging another model to provide step-level feedback to an actor model in the reasoning process (Xi et al., 2024; Welleck et al., 2023; Paul et al., 2024; Zhang et al., 2024e; Setlur et al., 2024; Zhang et al., 2024c; Guan et al., 2025; Zhang et al., 2024d), but such two-player frameworks suffer from high costs for model deployment. The most related to our work is SoS (Gandhi et al., 2024), which attempted to train a single LLM to perform a tree search as a flattened string. However, the effectiveness of SoS has primarily been demonstrated on simple symbolic tasks, and the ability to generalize to more complex problems, such as math word problems, remains to be explored.

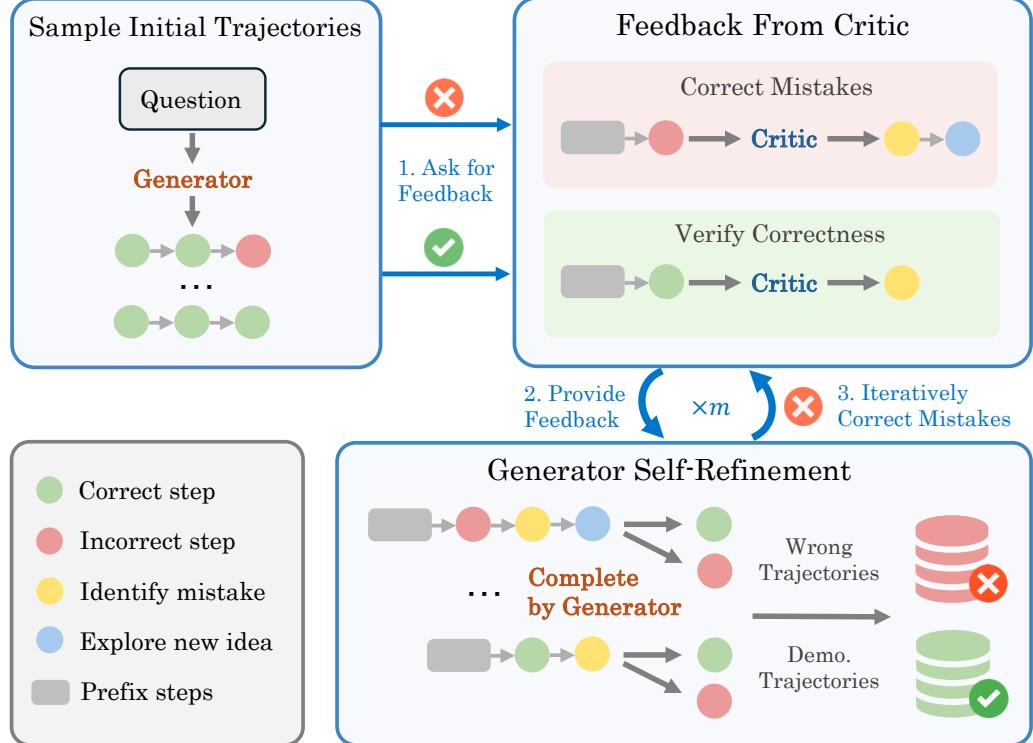

*Figure 17:* **Demonstration Trajectories Synthesis.** First, multiple initial reasoning trajectories are sampled from the generator and sent to critic to ask for feedback. The critic model identifies the mistake for trajectories with incorrect final answers and proposes an alternative solution. For trajectories with correct final answers, the critic model provides verification of its correctness. Based on the feedback, the generator self-refines its previous trajectories, and the incorrect trajectories are sent to the critic again for additional feedback with maximum $m$ iterations. At each step, those trajectories with successful refinements are preserved and finally, a reward model rates and collects high-quality demonstration trajectories to form the synthetic dataset $\mathcal{D}_{\text{syn}}$.

## C. Details about Data Synthesis Framework

**Sample Initial Trajectories.** The details of data synthesis framework are illustrated in Figure 17. Given an input problem $\boldsymbol{x} \in \mathcal{D}$, we begin by sampling the generator $\pi_g$ to generate $K$ initial reasoning trajectories. For each trajectory $\tilde{\boldsymbol{y}} = [\tilde{\boldsymbol{y}}_1, \tilde{\boldsymbol{y}}_2, \ldots, \tilde{\boldsymbol{y}}_L] \sim \pi_g(\cdot|\boldsymbol{x})$, we evaluate whether the final answer $\tilde{\boldsymbol{y}}_L$ matches the ground-truth answer $\boldsymbol{y}^*$. Based on the evaluation, the generated trajectories are divided into two subsets according to their correctness, which are then processed differently in subsequent steps.

**Critic and Refinement.** For those incorrect trajectories, the critic $\pi_c$ provides feedback to help the generator address its flaws. Specifically, the critic, given the ground-truth solution, identifies the first erroneous step $\tilde{\boldsymbol{y}}_l$ and generates a summary $\tilde{\boldsymbol{y}}_{l+1}$ of the mistake as a reflection, along with a exploration direction (hint), $\tilde{\boldsymbol{y}}_{l+2}$, i.e., $[\tilde{\boldsymbol{y}}_{l+1}, \tilde{\boldsymbol{y}}_{l+2}] \sim \pi_c(\cdot|\boldsymbol{x}, \tilde{\boldsymbol{y}}_1, \ldots, \tilde{\boldsymbol{y}}_l; \boldsymbol{y}^*)$. Next, we ask the generator $\pi_g$ to self-refine its current trajectory based on the feedback provided by the critic, performing a conditional generation of the remaining reasoning steps, $[\tilde{\boldsymbol{y}}_{l+3}, \ldots, \tilde{\boldsymbol{y}}_L] \sim \pi_g(\cdot|\boldsymbol{x}, \tilde{\boldsymbol{y}}_1, \ldots, \tilde{\boldsymbol{y}}_l; \tilde{\boldsymbol{y}}_{l+1}, \tilde{\boldsymbol{y}}_{l+2})$.

For correct trajectories, the critic $\pi_c$ focuses on verifying the correctness of the generator's reasoning. A random intermediate reasoning step $\tilde{\boldsymbol{y}}_l$ is selected, and the critic provides a summary explaining why the preceding steps are progressing toward the correct solution, i.e., $\tilde{\boldsymbol{y}}_{l+1} \sim \pi_c(\cdot|\boldsymbol{x}, \tilde{\boldsymbol{y}}_1, \ldots, \tilde{\boldsymbol{y}}_l; \boldsymbol{y}^*)$, where $\tilde{\boldsymbol{y}}_{l+1}$. Similarly, the generator continues from the current solution, generating the subsequent steps as $[\tilde{\boldsymbol{y}}_{l+2}, \ldots, \tilde{\boldsymbol{y}}_L] \sim \pi_g(\cdot|\boldsymbol{x}, \tilde{\boldsymbol{y}}_1, \ldots, \tilde{\boldsymbol{y}}_l; \tilde{\boldsymbol{y}}_{l+1})$.

Finally, we check whether the final answer $\tilde{\boldsymbol{y}}_L$ aligns with $\boldsymbol{y}^*$. The above procedure is repeated iteratively, up to a maximum of $m$ iterations, until the generator produces the correct final answer. All feedback and refinements are then contaminated to synthesize the final demonstration trajectories. Additionally, the trajectories are post-processed by inserting meta-action tokens at the beginning of each reasoning step to indicate its meta-action type.

**Trajectory Filtering.** The above procedure may yield multiple demonstration trajectories for each problem $x$. We then select the top-$k$ ($k < K$) trajectories based on the reward scores $r = \pi_r(\tilde{y}, x)$ assigned by the reward model $\pi_r$. This approach allows us to construct a diverse synthetic dataset $\mathcal{D}_{\text{syn}}$ containing high-quality demonstration trajectories, including (1) short-cut COAT paths that boil down CoT reasoning paths and (2) more complex COAT paths involving multiple rounds of self-reflection and exploration of alternative solutions.

# D. Experimental Setup

## D.1. Data Processing

**Data Source.** We construct our training dataset by combining two open-source synthetic datasets: OpenMathInstruct2 (Toshniwal et al., 2024a) and NuminaMath-COT (LI et al., 2024). After a careful review of the synthetic data, we identify and remove invalid questions to improve data reliability. To further enhance the quality of the dataset, we adopt the mutual consistency filtering method inspired by rStar (Qi et al., 2024a), which removes examples with inconsistent answers provided by different models. Specifically, we utilize QwQ (Qwen, 2024) to relabel the questions and compared the newly generated answers with the original answers from the source datasets. Only examples with consistent answers were retained. Additionally, we apply de-duplication tools from (Kocetkov et al., 2023) to eliminate redundant examples. Through these filtering processes, we finalized a high-quality dataset with approximately 550K samples in total.

**Multi-agent COAT Data Synthesis.** For the multi-agent demonstration trajectory synthesis framework, we utilize three models: Qwen-2.5-Math-7B-Instruct as the generator, Llama-3.1-70B-Instruct as the critic, and Skywork-Reward-Llama-3.1-8B-v0.2 as the outcome reward model. For the generator, we set the temperature to 0.3 and the maximum generation token limit to 2048. First, the generator samples $K = 100$ initial solutions for each problem, dividing the generated solutions into three subsets: correct, incorrect, and invalid (those that fail to produce a final answer). Invalid solutions are discarded, and we randomly select four correct solutions and four incorrect solutions. Next, we set sampling temperature of critic to 0.2 and a maximum token limit of 256, and let critic provides feedback on these selected solutions, allowing the generator to perform conditional generation. This iterative process is repeated for a maximum of $m = 2$ iterations, potentially resulting in up to $8 \times 2 = 16$ demonstration trajectories.

During the generation process, various situations require different prompt templates:

1. The generator produces an initial solution.

2. The initial solution is correct, and the critic verifies its correctness.

3. The initial solution is incorrect, and the critic identifies mistakes.

4. The generator generates continuations after the critic verifies the correctness of its correct initial solution.

5. The generator generates continuations after the critic identifies mistakes in its incorrect initial solution.

6. The generator fails to solve the problem after refinement, and the critic provides an additional feedback to identify errors in the generator's second attempt.

The prompt templates for these situations are detailed in Appendix D.1.1.

Among the synthetic trajectories, we categorize them into the following types:

1. **Type-I:** Synthetic trajectories without critic feedback, i.e., no reflection actions.

2. **Type-II-I:** Synthetic trajectories that include an intermediate reflection action to verify the correctness of previous reasoning steps.

3. **Type-II-II:** Synthetic trajectories that include 1) an intermediate reflection action to verify the correctness of previous reasoning steps, and 2) a second reflection action to correct mistakes in the previous solution, followed by an explore action to propose an alternative solution.

4. **Type-III-I:** Synthetic trajectories that include a reflection action to correct mistakes in the previous solution and an explore action to propose an alternative solution.

5. **Type-III-II:** Synthetic trajectories that include two rounds of self-reflection and self-explore.

Examples of these five types of synthetic trajectories are provided in Appendix D.1.2. Finally, the outcome reward model is applied to select the top-1 (k=1) sample of each type from the 16 demonstration trajectories based on the reward score, if such a type exists.

## D.1.1. PROMPT TEMPLATES

---

**Prompt Template 1.1 — Generator generates initial solution**

```
<|im_start|>user
Solve the following math problem efficiently and clearly.
Please reason step by step, and put your final answer within \boxed{}.
Problem: <<<instruction>>><|im_end|>
<|im_start|>assistant
```

---

**Prompt Template 1.2.1 — Generator generates continuations for correct partial solutions**

```
<|im_start|>system
Your task is to continue writing a solution to a problem. Given a problem, a correct partial solution, along with a sanity check
of previous steps, you should continue the current solution to solve the problem. Think step by step using English and adhere
to the following format.

Step: [Brief explanation and calculations]

Step: [Brief explanation and calculations]

...

**Please do not use other language than English.**

Your solution should conclude with "Therefore, the final answer is: \(\boxed{answer}\)", where [answer] is the final number
or expression that solves the problem. All steps in the solutions should be brief and always start with "Step:" and end with two
line breaks. Each subsequent step should not explicitly refer to previous steps, and the steps should constitute a coherent
progression towards the final answer.<|im_end|>
<|im_start|>user
## Problem
$$\frac{16^m}{4^n}=2$$
Find $m+n$.

## Partial solution with a sanity check

Step: First, we express 16 and 4 as powers of 2:
\[ 16 = 2^4 \]
\[ 4 = 2^2 \]

Step: Substitute these expressions into the given equation:
\[ \frac{(2^4)^m}{(2^2)^n} = 2 \]

Step: Use the power of a power property \((a^m)^n = a^{mn}\) to simplify the exponents:
\[ \frac{2^{4m}}{2^{2n}} = 2 \]

Step: Use the property of exponents \(\frac{a^m}{a^n} = a^{m-n}\) to combine the exponents:
\[ 2^{4m-2n} = 2^1 \]

Step: Since the bases are the same, we can set the exponents equal to each other:
\[ 4m - 2n = 1 \]
I correctly simplified the exponents and equated \(4m - 2n\) to 1, which is essential to find the relationship between \(m\)
and \(n\).<|im_end|>
<|im_start|>assistant
## Continued solution

Step: To find \( m + n \), we need to solve this equation for \( m \) and \( n \). First, divide the entire equation by 2 to
simplify: \[ 2m - n = \frac{1}{2} \]

Step: Add \( n \) to both sides to isolate \( 2m \):
\[ 2m = n + \frac{1}{2} \]

Step: Add \( m \) to both sides to find \( m + n \):
\[ m + n = \frac{1}{2} + \frac{1}{2} \]

Step: Simplify the right side:
\[ m + n = 1 \]

Therefore, the value of \( m + n \) is:
\[ \boxed{1} \]<|im_end|>
<|im_start|>user
## Problem
<<<instruction>>>

## Partial solution with a sanity check

<<<trajectory>>><|im_end|>
<|im_start|>assistant
## Continued solution

Step:
```

---

## Prompt Template 1.2.2 — Generator generates continuations for incorrect partial solutions

```
<|im_start|>system
Your task is to continue writing a solution to a problem. Given a problem, an incorrect partial solution where the last step is wrong, along with a revision feedback and a hint for next steps, you should continue the solution using the hint. Do not directly copy contents under "## Incorrect partial solution with revision feedback and hint for next steps". Think step by step using English and adhere to the following format.

Step: [Brief explanation and calculations]

Step: [Brief explanation and calculations]

...

**Please do not use other language than English.**

Your solution should conclude with "Therefore, the final answer is: \(\boxed{answer}\)", where [answer] is the final number or expression that solves the problem. All steps in the solutions should be brief and always start with "Step:" and end with two line breaks. Each subsequent step should not explicitly refer to previous steps, and the steps should constitute a coherent progression towards the final answer.<|im_end|>
<|im_start|>user
## Problem
Devanti wears her hair in various styles. A sleek bun is tied in top of the head, a half on the other with a long extension hanging down to the shoulders. To prepare for his next photo shoot, Devanti decides to ward off a few days of the daily grays until it reaches there full potential. Research shows that one-fourth of the gray hairs are deposited on the first day, and the second day is when half the remaining hair has disappeared, leaving Devanti with 15 gray hairs on her full head. Calculate the total number of the gray hairs that Devanti was able to tadpoles from her hair.

## Incorrect partial solution with revision feedback and hint for next steps

Step: Let \( x \) be the total number of gray hairs Devanti had initially.

Step: On the first day, one-fourth of the gray hairs are deposited on the head. Therefore, the number of gray hairs remaining after the first day is:
\[
x - \frac{1}{4}x = \frac{3}{4}x
\]
Wait. I subtracted one-fourth of x instead of adding it, but gray hairs are deposited (added) on the first day. Alternatively, I should add one-fourth of x to find the total number of gray hairs after the first day.<|im_end|>
<|im_start|>assistant
## Continued solution

Step: Let me correct the first day's calculation by adding one-fourth of the gray hairs.

\[
\text{Initial gray hairs} = x
\]
\[
\text{Gray hairs deposited on the first day} = \frac{1}{4}x
\]
\[
\text{Total gray hairs after the first day} = x + \frac{1}{4}x = \frac{5}{4}x
\]

Step: Then, calculate the number of gray hairs remaining after the second day, where half of them disappear.

\[
\text{Gray hairs after the second day} = \frac{1}{2} \times \frac{5}{4}x = \frac{5}{8}x
\]

Step: Next, set up the equation based on the information that 15 gray hairs remain after the second day.

\[
\frac{5}{8}x = 15
\]

Step: Finally, let's solve for \( x \) to find the initial number of gray hairs.

\[
x = 15 \times \frac{8}{5} = 24
\]

Therefore, the final answer is: \(\boxed{24}\).<|im_end|>
<|im_start|>user
## Problem
<<<instruction>>>

## Incorrect partial solution with revision feedback and hint for next steps

<<<trajectory>>><|im_end|>
<|im_start|>assistant
## Continued solution

Step:
```

---

### Prompt Template 2.1 — Critic verifies correctness

```
## General Guidelines
You are a student. Your task is to carefully review your own correct partial solution to a math problem, and adhere to the
following guidelines:

1. Verify the correctness of your own solution and explain your reason: "Verify: [brief explanation of why you are correct
with one sentence]"

3. You are provided with the question, the ground truth solution, and your step-by-step partial solution.

4. Your response should not include phrases like "ground truth solution".

5. Your response should be exactly in the following format:
Verify: [brief explanation of why you are correct
with one sentence]

## Test Example
### Question
<<<question>>>

### Ground truth solution
<<<gt_solution>>>

### Your partial solution
<<<student_solution>>>

### Your review
```

---

### Prompt Template 2.2.1 — Critic corrects mistakes

```
## General Guidelines
You are a student. Your task is to carefully review your own solution to a math problem, and adhere to the following guidelines:

1. Directly point out the first potentially incorrect step you find and explain your reason: "In Step <id>: [brief explanation
of the mistake with one sentence]"

2. After this, suggest an alternative step that you should have taken to correct the currect incorrect step: "Alternatively:
[your suggested step with one sentence]"

3. You are provided with the question, the ground truth solution, and your step-by-step solution.

4. The alternative step you propose should not include phrases like "ground truth solution".

5. Your response should be exactly in the following format:
In Step <id>: [brief explanation of the mistake in this step, with one sentence]

Alternatively: [your suggested new step, with one sentence]

## Test Example
### Question
<<<question>>>

### Ground truth solution
<<<gt_solution>>>

### Your incorrect solution
<<<student_solution>>>

### Your review
```

---

## Prompt Template 2.2.2 — Critic corrects mistakes for a second round

```
You are a student in a math class. You are collaborating with a partner to solve math problems.
First, your partner provided a partial solution to a math problem and identified a mistake.
Then, you continue your partner's partial solution and finish solving the problem.
Now, your task is to carefully review your continued solution, adhering to the following guidelines.

## Guidelines
1. "Your partner's partial solution" includes your partner's partial solution to the problem. It might be long and
contain several substeps. "Your continued solution" includes your completion of the problem based on your partner's partial solution.

2. Your task is NOT to check the correctness of "Your partner's partial solution", but to directly point out the first potentially
incorrect step in "Your continued solution", and explain the reason why you made the mistake: "In Step <id>: [brief explanation of
the mistake in your own solution with one sentence]".

3. After this, suggest an alternative step that you should have taken to correct your incorrect step: "Alternatively:
[your suggested step with one sentence]"

4. You are provided with the question, the ground truth solution, and your partner's partial solution, and your continued solution.

5. The alternative step you propose SHOULD NOT explicitly include phrases like "ground truth solution", "as shown in the ground
truth solution", etc.

6. You should only focus on your own solution. DO NOT refer to any mistake in your partner's partial solution.

7. Your response should be exactly in the following format:
In Step <id>: [brief explanation of the mistake in your own solution with one sentence]

Alternatively: [your suggested new step with one sentence]

## Test Example
### Question
<<<question>>>

### Ground truth solution
<<<gt_solution>>>

### Your partner's partial solution
<<<partner_solution>>>

### Your continued solution
<<<student_solution>>>

### Your review
```

## D.1.2. SYNTHETIC DATA EXAMPLES

### Synthetic Data Example Type I: No reflection

**[Input]**

Five couples plan to take their boats on a trip down a river. Each boat can carry 4 people. What is the smallest number of boats that can accommodate all five couples?

---

**[Output]**

First, we determine the total number of people. Since there are five couples, and each couple consists of 2 people, the total number of people is:

$$5 \times 2 = 10$$<|continue|>
Next, we need to figure out how many boats are required to carry all 10 people. Each boat can carry 4 people. To find the minimum number of boats, we divide the total number of people by the capacity of each boat:

$$\frac{10}{4} = 2.5$$<|continue|>
Since we cannot have half a boat, we need to round up to the next whole number. Therefore, we need 3 boats to accommodate all 10 people.

To verify, let's consider how the people can be distributed among the boats:
- Boat 1: 4 people
- Boat 2: 4 people
- Boat 3: 2 people

All 10 people are accounted for, and no boat exceeds its capacity.

Thus, the smallest number of boats that can accommodate all five couples is:

$$\boxed{3}$$

---

### Synthetic Data Example Type II-I: Verify correctness

[Input]

A library has a total of $50$ books, each of which is categorized as a biography, novel, or self-help book. The library's patrons borrow $30$ books in one week, and the ratio of borrowed biographies to borrowed novels to borrowed self-help books is $2:3:5$. If the library has an equal number of biographies, novels, and self-help books, how many self-help books were borrowed in that week?

---

[Output]

First, we determine the number of biographies, novels, and self_help books in the library. Since the library has an equal number of each type of book and the total number of books is 50, we can divide 50 by 3:

\[ \text{Number of biographies} = \text{Number of novels} = \text{Number of self\_help books} = \frac{50}{3} \]

However, since the number of books must be an integer and the problem states that the library has an equal number of each type of book, we need to re-evaluate the problem statement. The problem states that the library has an equal number of biographies, novels, and self_help books, but the ratio of borrowed biographies to borrowed novels to borrowed self_help books is \(2:3:5\). This means the total number of books borrowed must be a multiple of \(2+3+5=10\). Since 30 books were borrowed, it is indeed a multiple of 10.

<|reflect|>
Let me stop here to verify. The calculation of the number of each type of book in the library is correct, but the focus should be on the ratio of borrowed books, which is consistent with the given total of 30 borrowed books.<|continue|>
Let \( b \), \( n \), and \( s \) represent the number of biographies, novels, and self_help books borrowed respectively. According to the given ratio \(2:3:5\), we can write:
\[ b = 2x \]
\[ n = 3x \]
\[ s = 5x \]<|continue|>
The total number of books borrowed is 30:
\[ b + n + s = 30 \]<|continue|>
Substitute the expressions for \( b \), \( n \), and \( s \) into the equation:
\[ 2x + 3x + 5x = 30 \]<|continue|>
Combine like terms:
\[ 10x = 30 \]<|continue|>
Solve for \( x \):
\[ x = \frac{30}{10} = 3 \]<|continue|>
Find the number of self_help books borrowed:
\[ s = 5x = 5 \times 3 = 15 \]

Therefore, the number of self_help books borrowed in that week is:
\[ \boxed{15} \]

---

### Synthetic Data Example Type II-II: Verify correctness then correct mistake

[Input]

Consider the set of all positive integers whose digits, when read from left to right, form a valid English word. For example, 325 is a member of this set because "three two five" is a valid English phrase. What is the largest member of this set that is also a perfect square?

---

[Output]

To find the largest member of the set of positive integers whose digits form a valid English word and is also a perfect square, we need to consider the constraints of the problem. The number must be a perfect square and its digits must form a valid English word.

<|reflect|>
I want to double-check my calculations. The step correctly identifies the two main constraints of the problem: the number must be a perfect square and its digits must form a valid English word.<|continue|>
We start by checking the largest perfect squares in descending order until we find one whose digits form a valid English word.
<|continue|>
Let's check \( 9801 \):
\[ 9801 = 99^2 \]
The digits are 9, 8, 0, 1. "Nine eight zero one" is a valid English phrase.

Since \( 9801 \) meets both conditions (it is a perfect square and its digits form a valid English word), it is the largest such number.

Therefore, the largest member of the set that is also a perfect square is:
\[ \boxed{9801} \]<|reflect|>
But wait. The digits 9, 8, 0, 1 do not form a valid English word because "zero" is not a valid word in this context.
<|explore|>
I will now explore another method. We should check the next largest perfect square, which is 961, and verify if its digits form a valid English word.<|continue|>
Let's check \( 961 \):
\[ 961 = 31^2 \]
The digits are 9, 6, 1. "Nine six one" is a valid English phrase.

Since \( 961 \) meets both conditions (it is a perfect square and its digits form a valid English word), it is the largest such number.

Therefore, the largest member of the set that is also a perfect square is:
\[ \boxed{961} \]

---

## Synthetic Data Example Type III-I: Correct mistake

[Input]

Given $A=\left\{x\in \mathbb{N}^* \mid -x^{2}+3x\geqslant 0\right\}$ and $B=\left\{x \mid \log_{\frac{1}{2}}x\leqslant 0\right\}$, then $A\cap B=$

A: $\left[3,+\infty \right)$

B: $\left[0,1\right]$

C: $\left[1,3\right]$

D: $\left\{1,2,3\right\}$

---

[Output]

First, we solve the inequality for set \( A \):
\[ -x^2 + 3x \geq 0 \]<|continue|>
Factor out the common term:
\[ x(-x + 3) \geq 0 \]<|continue|>
Determine the critical points by setting each factor to zero:
\[ x = 0 \quad \text{or} \quad -x + 3 = 0 \implies x = 3 \]<|continue|>
The critical points divide the real number line into three intervals: \( (-\infty, 0) \), \( (0, 3) \), and \( (3, \infty) \). We test a point from each interval in the inequality \( x(-x + 3) \geq 0 \).

- For \( x \in (-\infty, 0) \), choose \( x = -1 \):
  \[ (-1)(-1 + 3) = (-1)(2) = -2 \quad (\text{not} \geq 0) \]

- For \( x \in (0, 3) \), choose \( x = 1 \):
  \[ (1)(1 + 3) = (1)(2) = 2 \quad (\geq 0) \]

- For \( x \in (3, \infty) \), choose \( x = 4 \):
  \[ (4)(4 - 3) = (4)(1) = 4 \quad (\geq 0) \]<|continue|>
 Since the inequality is non-strict (\(\geq\)), we include the critical points \( x = 0 \) and \( x = 3 \). Therefore, the
 solution to the inequality is:
\[ A = [0, 3] \]<|continue|>
Next, we solve the inequality for set \( B \):
\[ \log_{\frac{1}{2}} x \leq 0 \]<|continue|>
Recall that the base of the logarithm is \(\frac{1}{2}\), which is less than 1. For a logarithm with a base less than 1, the
inequality \(\log_{\frac{1}{2}} x \leq 0)\) implies:
\[ x \geq 1 \]

<|reflect|>
I may have overlooked something. The solution to the inequality incorrectly includes the critical point \( x = 0 \) because \( x
\in \mathbb{N}^* \) (the set of positive natural numbers), and \( 0 \) is not a positive natural number.<|explore|>
To find a better solution, The solution to the inequality should be \( A = \left\{1, 2, 3\right\} \) since \( x \) must be
a natural number and \( 0 < x \leqslant 3 \).<|continue|>
Correct the solution for set \( A \) considering \( x \in \mathbb{N}^* \):

\[ A = \{1, 2, 3\} \]<|continue|>
Set \( B \) remains the same:

\[ B = [1, \infty) \]<|continue|>
Find the intersection \( A \cap B \):

\[ A \cap B = \{1, 2, 3\} \cap [1, \infty) = \{1, 2, 3\} \]

Therefore, the final answer is:
\[ \boxed{D} \]

---

### Synthetic Data Example Type III-II: Correct mistake multiple times

[Input]

In $\triangle ABC$, $a$, $b$, $c$ are the sides opposite to the angles $A$, $B$, $C$ respectively. Given $c=4$, $\tan A=3$, $\cos C= \frac{ \sqrt{5}}{5} $, find the area of $\triangle ABC$.

---

[Output]

First, we find $\sin C$ using the Pythagorean identity $\sin^2 C + \cos^2 C = 1$:

$$\sin^2 C = 1 - \cos^2 C = 1 - \left(\frac{\sqrt{5}}{5}\right)^2 = 1 - \frac{5}{25} = \frac{20}{25} = \frac{4}{5}$$

Since $C$ is an angle in a triangle, $\sin C$ is positive:

$$\sin C = \frac{2}{5}$$<|continue|>
Next, we find $\sin A$ using the given $\tan A = 3$. Recall that $\tan A = \frac{\sin A}{\cos A}$. Let $\sin A = 3k$ and $\cos A = k$. Using the Pythagorean identity $\sin^2 A + \cos^2 A = 1$:

$$(3k)^2 + k^2 = 1 \implies 9k^2 + k^2 = 1 \implies 10k^2 = 1 \implies k^2 = \frac{1}{10} \implies k = \frac{1}{\sqrt{10}} = \frac{\sqrt{10}}{10}$$

Thus, $\sin A = 3k = 3 \cdot \frac{\sqrt{10}}{10} = \frac{3\sqrt{10}}{10}$ and $\cos A = k = \frac{\sqrt{10}}{10}$.<|continue|>
We use the Law of Sines to find $a$:

$$\frac{a}{\sin A} = \frac{c}{\sin C} \implies \frac{a}{\frac{3\sqrt{10}}{10}} = \frac{4}{\frac{2}{5}} \implies \frac{a}{\frac{3\sqrt{10}}{10}} = 4 \cdot \frac{5}{2} = 10 \implies a = 10 \cdot \frac{3\sqrt{10}}{10} = 3\sqrt{10}$$

<|reflect|>
Wait a minute. The calculation of $\sin C$ is incorrect because $\sin C$ should be $\frac{2}{\sqrt{5}}$ instead of $\frac{2}{5}$.<|explore|>
This time, Use the correct value of $\sin C = \frac{2}{\sqrt{5}}$ in subsequent calculations.<|continue|>
Correct the calculation for $\sin C$:

$$\sin C = \frac{2}{\sqrt{5}}$$<|continue|>
Use the correct value of $\sin C$ in the Law of Sines to find $a$:

$$\frac{a}{\sin A} = \frac{c}{\sin C} \implies \frac{a}{\frac{3\sqrt{10}}{10}} = \frac{4}{\frac{2}{\sqrt{5}}} \implies \frac{a}{\frac{3\sqrt{10}}{10}} = 4 \cdot \frac{\sqrt{5}}{2} = 2\sqrt{5} \implies a = 2\sqrt{5} \cdot \frac{3\sqrt{10}}{10} = \frac{6\sqrt{50}}{10} = \frac{6 \cdot 5\sqrt{2}}{10} = 3\sqrt{2}$$<|continue|>
 Use the formula for the area of a triangle $( \text{Area} = \frac{1}{2}ab \sin C )$:

$$\text{Area} = \frac{1}{2} \cdot 3\sqrt{2} \cdot 4 \cdot \frac{2}{\sqrt{5}} = \frac{1}{2} \cdot 12\sqrt{2} \cdot \frac{2}{\sqrt{5}} = \frac{1}{2} \cdot \frac{24\sqrt{2}}{\sqrt{5}} = 12 \cdot \frac{\sqrt{2}}{\sqrt{5}} = 12 \cdot \sqrt{\frac{2}{5}} = 12 \cdot \frac{\sqrt{10}}{5} = \frac{12\sqrt{10}}{5}$$

Therefore, the area of $\triangle ABC$ is: $\boxed{\frac{12\sqrt{10}}{5}}$.<|reflect|>
Let's reassess my previous conclusion. The mistake is in the calculation of the area, where the value of $\sin B$ should have been used instead of $\sin C$.<|explore|>
To find a better solution, Use the correct value of $\sin B = \frac{\sqrt{2}}{2}$ in the area formula: $\text{Area} = \frac{1}{2} \cdot 3\sqrt{2} \cdot 4 \cdot \frac{\sqrt{2}}{2} = 6$.<|continue|>
correct the calculation of the area using the correct value of $\sin B = \frac{\sqrt{2}}{2}$: $\text{ Area} = \frac{1}{2} \cdot 3\sqrt{2} \cdot 4 \cdot \frac{\sqrt{2}}{2} = 6$.

Therefore, the area of $\triangle ABC$ is: $\boxed{6}$.

## D.2. Format Tuning

**Training Details.** We perform supervised fine-tuning (SFT) on the Qwen-2.5-Math-7B base model using our synthetic COAT dataset with 10K unique questions. The SFT prompt template is illustrated below. We utilize a cosine learning rate scheduler with an initial learning rate of 2e-5. The batch size is set to 128, the maximum sequence length is 4096, and the model is trained for a maximum of two epochs. We add the following special tokens `<|continue|>`,`<|reflect|>`,`<|explore|>` into the vocabulary. All experiments are implemented using the LLaMA-Factory framework (Zheng et al., 2024).

---
**Prompt Template 3: SFT and RL**

```
<|im_start|>user
Solve the following math problem efficiently and clearly.
Please reason step by step, and put your final answerwithin \boxed{}.
Problem: <<<your instruction>>>
<|im_start|>assistant
```
---

## D.3. Reinforcement Learning

**ORM Training.** To construct the preference data for our ORM models, we utilize our format-tuned model, Satori-Qwen-7B-FT, to generate trajectories. Starting with our filtered training dataset of 550K unique questions, we follow these steps: (1) allow the FT model to sample eight solutions for each question; (2) evaluate the correctness of these solutions and label them accordingly; and (3) select only those questions that contain correct and incorrect solutions. For these selected questions, we construct preference data by pairing correct solutions with their corresponding incorrect ones, resulting in a preference dataset of approximately 300K unique questions.

For each problem $x$, we allow $\pi_\theta$ to randomly generate multiple reasoning trajectories, constructing a dataset $\mathcal{D}_r$ with positive and negative pairs of trajectories. We select trajectories with the correct final answer as positive trajectories $\tilde{y}^+$ and trajectories with incorrect final answer as negative trajectories $\tilde{y}^-$. Assuming the Bradley-Terry (BT) preference model, we optimize the reward model $r_\psi(x, \tilde{y})$ through negative log-likelihood,

$$\mathcal{L}_{rm}(\psi) := -\mathbb{E}_{(\boldsymbol{x},\tilde{\boldsymbol{y}}^+,\tilde{\boldsymbol{y}}^-)\sim\mathcal{D}_r} \left[ \log\left(\sigma\left(r_\psi(\boldsymbol{x},\tilde{\boldsymbol{y}}^+) - r_\psi(x,\tilde{\boldsymbol{y}}^-) - \tau\right)\right)\right]$$

where $\tau$ denotes a target reward margin. In practice, we observe that setting $\tau > 0$ improves the performance of the reward model.

**RL Data.** Our RL training dataset consists of 300K unique questions from the preference dataset. This ensures that the questions are neither too easy (where the FT model always produces correct solutions) nor too difficult (where the FT model never succeeds). This encourages policy to learn through trial and error during RL training. To further guide the model to learn self-reflection capabilities, we apply the proposed RAE technique, augmenting input problems with restart buffers, i.e., intermediate reasoning steps collected from the FT model. These intermediate steps are extracted from the preference dataset, and for each question, we randomly select one correct and one incorrect trajectory, applying the back-track technique for up to $T = 2$ steps.

**Training Details.** For both ORM and RL training, we implement our experiments using the OpenRLHF framework (Hu et al., 2024). For ORM training, we employ a cosine learning rate scheduler with an initial learning rate of 2e-6. The batch size is set to 128, the maximum sequence length to 4096, and the model is trained for two epochs. As the objective function, we use PairWiseLoss (Christiano et al., 2017) with a margin of $\tau = 2$. For evaluation, we select the optimal ORM model checkpoint based on RM@8 performance, measured using the SFT model on a held-out validation dataset. Specifically, we allow the FT model to sample eight trajectories and let ORM select the best trajectory according to the highest reward score. The RM@8 accuracy is then computed based on the selected trajectories.

For RL training, we use the PPO algorithm (Schulman et al., 2017a). The critic model is initialized from our ORM model, while the actor model is initialized from our FT model. We optimize the models using a cosine learning rate scheduler, setting the learning rate to 2e-7 for the actor model and 5e-6 for the critic model. During PPO training, we sample one trajectory per prompt. The training batch size is set to 128, while the rollout batch size is 1024. Both the number of epochs

and episodes are set to 1. The maximum sequence length for prompts and generations is fixed at 2048. Additional parameter settings include a KL coefficient of 0.0, a sampling temperature of 0.6, and a bonus scale of $r_{\text{bonus}} = 0.5$.

**Second-round Self-improvement.** We begin with a set of 240K unique questions, also used in the distillation experiments shown in Table 5. The policy of the first round of RL training serves as a teacher model to generate synthetic reasoning trajectories. Among these 240K questions and corresponding trajectories, we filter the data based on question difficulty, selecting the most challenging 180K samples for distillation. This process results in a new fine-tuned (FT) model checkpoint, obtained from supervised fine-tuning (SFT) on these 180K trajectories. Since the new FT model has been trained on numerous high-quality trajectories, including reflection actions distilled from the teacher model, we do not apply restart and exploration (RAE) techniques in the second round of RL training to further encourage reflection. Additionally, we increase the sampling temperature from 0.6 to 1.2, generating eight samples per prompt to encourage more aggressive exploration to push the performance limit.

### D.4. Evaluation Details.

For each model, we use the same zero-shot CoT prompt template to obtain results on all test datasets. For Satori and all its variants, we use Prompt Template 3 (Appendix D.2). We set the temperature to 0 (greedy decoding) for every model, and collect pass@1 accuracies. Details of each test dataset are as follows.

**MATH500** (Lightman et al., 2023) is a subset of MATH (Hendrycks et al., 2021a) of uniformly sampled 500 test problems. The distribution of difficulty levels and subjects in MATH500 was shown to be representative of the entire MATH test set.

**GSM8K** (Cobbe et al., 2021) is a math dataset that consists of 8.5K high-quality, linguistically diverse grade-school math word problems designed for multi-step reasoning (2 to 8 steps). Solutions involve elementary arithmetic operations and require no concepts beyond early algebra. Its test set contains 1319 unique problems.

**OlympiadBench** (He et al., 2024) is a bilingual, multimodal scientific benchmark with 8,476 Olympiad-level math and physics problems, including those from the Chinese college entrance exam. We use the open-ended, text-only math competition subset, containing 674 problems in total.

**AMC2023** and **AIME2024** contain 40 text-only problems from American Mathematics Competitions 2023 and 30 text-only problems from American Invitational Mathematics Examination 2024, respectively.

**FOLIO** (Han et al., 2024) is a human-annotated dataset designed to evaluate complex logical reasoning in natural language, featuring 1,430 unique conclusions paired with 487 sets of premises, all validated with first-order logic (FOL) annotations. Its test set contains 203 unique problems.

**BoardgameQA (BGQA)** (Kazemi et al., 2024) is a logical reasoning dataset designed to evaluate language models' ability to reason with contradictory information using defeasible reasoning, where conflicts are resolved based on source preferences (e.g., credibility or recency). Its test set contains 15K unique problems.

**CRUXEval** (Gu et al., 2024) is a benchmark for evaluating code reasoning, understanding, and execution, featuring 800 Python functions (3-13 lines) with input-output pairs for input and output prediction tasks. Given a function snippet and an input example, LLMs are tasked to generate the corresponding outputs. Its test set contains 800 unique problems.

**StrategyQA** (Geva et al., 2021) is a question-answering benchmark designed for multi-hop reasoning where the necessary reasoning steps are implicit and must be inferred using a strategy. Each of the 2,780 examples includes a strategy question, its step-by-step decomposition, and supporting Wikipedia evidence.

**TableBench** (Wu et al., 2024a) is a tabular reasoning benchmark designed to evaluate LLMs on real-world tabular data challenges, covering 18 fields across four major TableQA categories: Fact checking, numerical reasoning, data analysis, and code generation for visualization. We test all models on fact checking and numerical reasoning subsets for simplicity of answer validation, resulting in 491 unique problems.

**MMLUProSTEM** is a subset of MMLU-Pro (Wang et al., 2024b). MMLU-Pro is an enhanced benchmark designed to extend MMLU (Hendrycks et al., 2021b) by incorporating more reasoning-focused questions, expanding answer choices from four to ten, and removing trivial or noisy items. We select six STEM subsets: physics, chemistry, computer science, engineering, biology, and economics (we remove the math subset as it belongs to in-domain tasks). Finally, we obtain 5371 unique problems in total.

# E. Additional Results

## E.1. Ablation on Reflection Bonus

*Table 6:* **Ablation Study on Reflection Bonus.**

| Bonus Scale | GSM8K | MATH500 | Olym. | AMC2023 | AIME2024 |
|---|---|---|---|---|---|
| **0.0** | 93.6 | 84.4 | 48.9 | 62.5 | 16.7 |
| **0.5 (default)** | 93.2 | 85.6 | 46.6 | 67.5 | 20.0 |

During RL training, we introduce a reflection bonus to facilitate the policy to learn self-reflection capabilities. The default value of the reflection bonus is set to $r_{\text{reflect}} = 0.5$. To analyze its impact on performance, we also evaluate the model with the reflection bonus set to $r_{\text{reflect}} = 0$. The results are presented in Table 6. We observe that the performance slightly degrades on challenging benchmark AMC2023 and AIME2024 when set $r_{\text{reflect}} = 0$ compared to $r_{\text{reflect}} = 0.5$.

## E.2. Ablation on Restart and Explore (RAE)

*Table 7:* **Ablation Study on Restart and Explore (RAE).**

| Setting | GSM8K | MATH500 | Olym. | AMC2023 | AIME2024 |
|---|---|---|---|---|---|
| **No RAE** | 93.4 | 81.6 | 45.2 | 57.5 | 20.0 |
| **with RAE** | 93.2 | 85.6 | 46.6 | 67.5 | 20.0 |

Complementary to the reflection bonus, the RAE technique effectively encourages the model to self-reflect and improves the diversity of initial states. We conduct an ablation study by removing the RAE technique in RL training. As shown in Table 7, the model without RAE exhibits performance degradation, confirming that RAE plays a critical role in our RL training.

## E.3. Ablation on Preference Bonus

*Table 8:* **Ablation Study on Preference Bonus.**

| Reward | GSM8K | MATH500 | Olym. | AMC2023 | AIME2024 |
|---|---|---|---|---|---|
| **Only Rule-based** | 93.6 | 82.8 | 46.3 | 62.5 | 16.7 |
| **ORM + Rule-based** | 93.2 | 85.6 | 46.6 | 67.5 | 20.0 |

To mitigate the sparse reward problem, we incorporate a preference bonus provided by an Outcome Reward Model (ORM). To assess the impact of the preference bonus, we perform an ablation study by removing the ORM and relying solely on the rule-based reward and reflection bonus as the RL reward signal. The results in Table 8 demonstrate performance degradation, suggesting the effectiveness of using a hybrid reward signal that combines sparse rule-based rewards with dense ORM-based rewards.

## E.4. Offline Restart Buffer v.s. Online Restart Buffer

Complementary to the reflection bonus, the restart buffer is designed to enhance the policy's self-reflection capabilities by augmenting the initial states with a diverse set of intermediate states. This includes trajectories processed from both correct and incorrect reasoning paths, which are then categorized into positive ($\mathcal{D}_{\text{restart}}^{+}$) and negative ($\mathcal{D}_{\text{restart}}^{-}$) restart buffers, as described in Section 4.2.

In addition to constructing the restart buffer offline, we also explore an online restart buffer approach. Specifically, after each PPO episode, we use the updated policy to construct the restart buffer and collect rollouts from this buffer to optimize the policy, iteratively repeating this process. However, this approach is suboptimal. During PPO training, we observe that the majority of sampled trajectories are correct, leading to a significant imbalance between correct and incorrect intermediate states in the online restart buffer. As a result, the model fail to adequately learn from incorrect paths, which are essential for incentivize self-reflection actions.

To overcome this limitation, we opt for an offline restart buffer approach to mitigate the bias introduced by online collection. Offline sampling ensures a balanced inclusion of intermediate states from both correct and incorrect trajectories.

