# OpenReview forum: "Satori: Reinforcement Learning with Chain-of-Action-Thought Enhances LLM Reasoning via Autoregressive Search"
_ICML.cc/2025/Conference — ICML 2025 poster_

### Official Review · Reviewer_X1zu · 2025-03-11

**Overall Recommendation:** 2

**Summary:**

This paper propose a post-training framework, incluing two stages: (1)  the chain-of-Action-thought fine-tuning, i.e., format fine-tuning; (2) self-improvement RL, i.e., iterative distillation and RL. For first stage, using Qwen-2.5-Math-Instruct and Llama-3.1-70B-Instruct constructs a multi-agent data synthesis framework. For second stage, incorpating rule-based  reward, reflection Bonuses and preference bonuses into final reward. Authors evaluate their resultsing model in GSM8K, MATH500, AMC2023, AIME2024, and OlympiadBench. They achieve comparable performance with other LLMs.

## Update

Thanks for detail reponse.  I summary my concern as follow:
1. Overclaim of performance. As shown in table 1, the propose method did not outperform than baselines. Since this paper have no theoretical analysis, authors should highlight the up/down and numbers comparing to baselines.
2. Need more statics on augmented data. The new format of RL data is the core contribution. For more better understanding, authors should provide ablation on data.
3. This paper is more like a technical report than an academic paper, including data format, integrated reward and so on. Authors should provide more insights for academic community.
Finally, i will keep my socre.

**Claims And Evidence:**

There are some over-claims:
1. "without external guidance" -> this paper adopt a a multi-agent system in data construction, including Qwen-2.5-Math-Instruct and Llama-3.1-70B-Instruct. If author want to illustrate the efficiency, I suggest they provide more evidence about cost, FLOPS or other metric of efficiency.
2. "superior performance" -> GSM8K: 93.9 (theirs) vs  95.2/91.6 (Qwen /w math). The same situation applies to MATH500. And, they also achieve 4 point improvement on average of OlympiadBench.

**Essential References Not Discussed:**

Not applicable.

**Experimental Designs Or Analyses:**

Yes, this paper's experimental designs are reasonable.

**Methods And Evaluation Criteria:**

Yes, this paper use reasonable metric for evaluation.

**Other Comments Or Suggestions:**

If the paper includes more analysis on reinforcement learning theory, it would provide deeper insights.

**Other Strengths And Weaknesses:**

The main concerns of this paper are :
1. there are not theoretical results.
2. the complex and diverse training system design ultimately only achieves comparable performance, and the results presented in the paper do not support the claims made in the introduction.

**Questions For Authors:**

- If you perform SFT using only your synthesized data in stage 1, would the performance differ from the current results?
- And is there an R1-style comparison?

**Relation To Broader Scientific Literature:**

The key difference between this paper and current literature is the fine-tuning data format and reward design in RL. They incorporate all suitable formats for self-improving fine-tuning. And for reward designs, they propose reflection bonuses and preference bonuses.

**Theoretical Claims:**

This paper does not provide any theoretical results or insights.

---

> ### Author Rebuttal · Authors · 2025-03-31
>
> **1. Over-claim about "without external guidance".**
>
> We would like to clarify that **“external guidance” specifically refers to guidance provided by another LLM verifier at inference time (see Abstract, lines 18-19)**. Many existing LLM reasoning methods rely on extensive sampling and guidance from a verifier model (e.g., PRM-guided tree search). Regarding the cost of the multi-agent system, we believe this is analogous to the inevitable cost of curating most LLM training datasets. We will clarify these points in the revision.
>
> **2. Over-claim about "superior performance".**
>
> We would like to highlight that the **performance gains on challenging MATH benchmarks are substantial**. For example, we achieved +5% on OlympiadBench, +10% on AMC2023, and +3.3% on AIME2024 compared to the best available 7B models. Moreover, **on our out-of-domain generalization experiments**, our method consistently outperforms baseline models **with an average improvement of 8.3% (see Table 2)**. While we acknowledge that the paper review may carry some degree of subjectivity, we would also like to note that **all three other reviewers appreciate the strong empirical performance**. However, we would like to tone down the performance claims in the revision.
>
> **3. No theoretical results.**
>
> We acknowledge the reviewer’s concern regarding the lack of theoretical analysis, as our proposed training framework currently does not come with a rigorous theoretical justification. However, we would like to note that conducting a comprehensive theoretical analysis in the context of LLM, especially for complex RL training pipelines is highly non-trivial and beyond the scope of this work. We also agree that some components, particularly the proposed Restart and Explore (RAE) technique, merit further investigation. We view this as an important direction for future work and include some preliminary theoretical insights about RAE as a starting point for such analysis.
>
>
> RAE modifies the initial state distribution by starting new rollouts not only from dataset-sampled prompts but also from random partial trajectories, i.e., prompts concatenated with partial responses generated by intermediate policies. This design can be justified by Theorem 6.2 of the Conservative Policy Iteration (CPI) paper [1], It states that if $\pi$ is an approximately optimal policy learned from one initial distribution $\mu$, and $\pi^*$ is the true optimal policy for a different initial distribution $\tilde{\mu}$, then
>
> $$
> \eta_{\tilde{\mu}}(\pi^*) - \eta_{\tilde{\mu}}(\pi)
> \ \le \
> \frac{\epsilon}{(1-\gamma)^2}
> \left\|\frac{d_{\pi^*, \tilde{\mu}}}{\mu}\right\|_\infty,
> $$
>
> where $d_{\pi^*, \tilde{\mu}}$ is the stationary state distribution of $\pi^*$ under $\tilde{\mu}$, $\gamma$ is the discount factor, and $\epsilon$ is small approximation error. The concentrability coefficient $\bigl\|\frac{d_{\pi^*, \tilde{\mu}}}{\mu}\bigr\|_\infty$ measures how badly $\mu$ misses states that $\pi^*$ visits. If $\mu$ puts little or no mass on those important states, the policy learned from $\mu$ can perform poorly under $\tilde{\mu}$.
>
>
> In the context of LLMs, the "initial state" corresponds to the prompt prefix. Training with a narrow or fixed set of prompts results in a limited $\mu$, which may place little to no mass on the key states an optimal policy should visit, thereby inflating the concentrability coefficient. RAE mitigates this issue by broadening $\mu$ through randomized partial rollouts, effectively augmenting the initial state distribution with more diverse and relevant prefixes encountered during training. While we do not have access to the true stationary distribution $d_{\pi^*, \tilde{\mu}}$, this augmentation strategy reduces distribution mismatch and narrows the theoretical performance gap between $\pi$ and $\pi^*$. This aligns with the conclusion in [1], which argues that more uniform state coverage leads to tighter bounds.
>
> We plan to explore a more formal and comprehensive analysis of RAE in future work.
>
>
> **4. Question-1: If you perform SFT using only your synthesized data in stage 1, would the performance differ from the current results?**
>
> **We kindly refer the reviewer to the “Large-scale FT vs. Large-scale RL” ablation study in Section 6**, where we have demonstrated that performing SFT using only the large-scale synthesized data in Stage 1 is sub-optimal compared to large-scale RL training.
>
> **5. Question-2: Is there an R1-style comparison?**
>
>  Please refer to [**our response to reviewer wWt8**](https://openreview.net/forum?id=j4FXxMiDjL&noteId=lC8KFYxRMy) for details.
>
> **Reference**
>
> - [1] Approximately optimal approximate reinforcement learning. ICML, 2002.
>
> ***
>
> **If these clarifications satisfactorily address the reviewer's concerns, we kindly ask if the reviewer would consider updating the score to reflect what we believe is a paper with noteworthy contributions to the community.**

---

### Official Review · Reviewer_WTyn · 2025-03-14

**Overall Recommendation:** 3

**Summary:**

The paper presents a method called Satori, which could enhance the reasoning abilities of LLMs. It does this through Chain-of-Action-Thought (COAT), a system that adds special “meta-action” tokens (like ``<|reflect|>`` and ``<|explore|>``) to regular chain-of-thought prompts. These tokens let the model pause to check its work or try a different approach on its own. The authors train the model in two main stages: first, they fine-tune it on a small set of example trajectories to teach it how to use the new tokens. Second, they apply RL at a much larger scale, which allows the model to generate new solutions, find and fix its mistakes, and gradually improve how well it reasons.

**Claims And Evidence:**

Yes. It is clear and convincing

**Essential References Not Discussed:**

No

**Experimental Designs Or Analyses:**

They evaluate performance on datasets including GSM8K, MATH, and several out-of-domain tasks. Their experiments validate their claim that Satori can both solve challenging math problems and generalize to new domains.

**Methods And Evaluation Criteria:**

Yes. It makes sense.

**Other Comments Or Suggestions:**

No other comments

**Other Strengths And Weaknesses:**

Their method is undeniably powerful, and they provide strong evidence for its effectiveness. However, it remains unclear whether this approach can surpass the latest open-sourced r1 model.

**Questions For Authors:**

In many RL settings, methods that rely heavily on pre-trained or trainable critic models can be easily hacked by LMs. How does your approach mitigate these risks? Could you explain any strategies you employ to ensure the policy cannot simply exploit the critic’s learned features?

**Relation To Broader Scientific Literature:**

I understand the authors mentioned r1 model in their paper. However, I still want to mention the following fact:

Because o1 and o3 models are not open-sourced, it is unclear how OpenAI trained them. However, DeepSeek’s r1 shows performance comparable to o1 and reportedly uses a cold-start reinforcement learning approach, which seems to let the language model develop reflection skills on its own (the “Aha moment”). In contrast, the paper in question relies on a specialized dataset to teach reflection-like abilities. Compared with r1’s algorithm, this approach has several constraints: (1) it requires format tuning prior to self-improvement, limiting its flexibility to learn other skills, and (2) this format tuning could negatively affect the model’s existing capabilities. Moreover, their final results still lag behind o1, suggesting that while their method works, it may be somewhat behind the latest approaches.

**Theoretical Claims:**

They make no theoretical claims

---

> ### Author Rebuttal · Authors · 2025-03-31
>
> **1. R1 uses a cold-start RL to develop reflection skills on its own (the “Aha moment”), but the paper relies on a specialized dataset to teach reflection-like abilities.**
>
> We would like to respectfully clarify several important points:
>
> - **R1 is a concurrent work and comparison should not be required:** r1 was officially released around the same time as the ICML 2025 abstract deadline (end of January). **According to the reviewer guidelines, we are not required to compare with or even mention r1.** However, we chose to acknowledge r1 as a concurrent work because we believe it offers valuable context for the research community amidst the recent interest it has generated.
> - **Clarifying potential misunderstanding:** We respectfully suggest that the reviewer may have some confusion regarding r1. (1) Deepseek introduces two models: r1-zero, which is trained using pure RL from a base model, and r1, the actual released model. R1 includes a SFT stage before RL, where the SFT data is explicitly designed with a reflection pattern (see Section 2.3.1 in [1]). Thus, **r1 employs a similar strategy to our format-tuning stage, although its implementation details are not disclosed**. (2) **Rather than limiting capabilities, this SFT stage is helpful to achieve more effective RL**. Indeed, the r1 authors acknowledge the limitations of r1-zero without SFT (see Section 2.2.4 in [1]), including poor readability and language mixing issues. Furthermore, recent work [2] supports the effectiveness of incorporating SFT before RL. Several studies have also questioned the validity of the “Aha moment” claimed by r1-Zero and suggest “Aha moment” could be a mirage (see [3]), which challenges the reviewer's claim that "r1 can develop reflection skills on its own".
> - **Comparison results provided:** Please refer to [**our response to reviewer wWt8**](https://openreview.net/forum?id=j4FXxMiDjL&noteId=lC8KFYxRMy).
>
> **2. The performance still lags behind o1 and open-sourced r1 model.**
> - **Unfair comparison:** We respectfully argue that it is not reasonable to compare the performance of a research prototype with that of large-scale industry systems. R1 contains over 600B parameters, whereas our model is based on a 7B LLM. Furthermore, the scale of training data and infra used by o1 and r1 is not publicly disclosed but is far beyond what is accessible in a academic setting.
> - **R1 is not truly open-sourced:** While o1 is certainly closed-source, we would also like to clarify that r1 is not fully open-sourced. Only the model weights have been released. The training data, codebase, and detailed pipeline remain proprietary. Over the past two months, several efforts have attempted to reproduce r1’s performance, but few—if any—have succeeded in matching it, which further suggests that important implementation details are missing.
> - **Research Objective:** o1 and r1 are undoubtedly successful, but their performance is largely attributed to extensive resources. In contrast, the goal of academic research is not to achieve SOTA benchmark results, but rather to explore novel methodologies orthogonal to scaling up resources. For instance, our proposed RAE technique effectively mitigates the sparse reward issue in RL, which remains a major challenge for complex reasoning tasks and is under-explored in prior works. Moreover, RAE could potentially be integrated into r1’s training framework as well. We respectfully note that if benchmark performance were the sole criterion for evaluation, then almost no academic work (including LLM reasoning papers submitted to ICML 2025) could surpass current industry-developed models such as r1 or o1.
>
> **3. How does your approach mitigate the risk of critic model hacking?**
> - **Request for clarification**: In RL, the term “hacking” typically refers to reward hacking, where the model exploits flaws in a reward model to receive undeservedly high scores. By contrast, the critic model in PPO is used for estimating advantages via GAE, and does not directly influence the reward signal. We kindly ask the reviewer to clarify whether the concern is regarding to reward model or PPO’s critic model.
> - **Our approach**: Satori utilizes a hybrid reward mechanism with both a reward model and rule-based reward. (see Section 4.2). While it is possible that the reward model may become less reliable during later stages of training, our design ensures that the rule-based reward remains dominant, mitigating the risk of reward hacking.
>
> **Reference**
> - [1] DeepSeek-R1: Incentivizing Reasoning Capability in LLMs via Reinforcement Learning, Jan, 2025.
> - [2] Demystifying Long Chain-of-Thought Reasoning in LLMs, Feb 2025.
> - [3] There May Not be Aha Moment in R1-Zero-like Training — A Pilot Study, Feb, 2025.
>
> ***
> **If these clarifications satisfactorily address the reviewer's concerns, we kindly ask if the reviewer would consider updating the score to reflect what we believe is a paper with noteworthy contributions to the community.**

---

> > ### Comment · Reviewer_WTyn · 2025-04-09
> >
> > Thank you for your thoughtful and detailed response. I now understand that r1 is a concurrent effort and therefore need not be included in the comparisons for this conference.
> >
> > I’d like to clarify one small point. You noted that “this SFT stage is helpful to achieve more effective RL,” but [1] reports that r1‑zero outperforms r1 on reasoning tasks, implying that format tuning can sometimes reduce model performance even after RL. I thought this nuance might be helpful to flag.
> >
> > After reviewing your explanation, I realize my earlier evaluation was too harsh, so I have raised my score accordingly. Your framework is indeed a fair and promising direction for future language‑model training.
> >
> > Thank you again for the engaging discussion.

---

### Official Review · Reviewer_wWt8 · 2025-03-14

**Overall Recommendation:** 4

**Summary:**

This work studies the problem of post-training LLMs for self-reflection and self-exploration capabilities. A training scheme termed COAT is proposed, which consists of two stages: (1) a small-scale SFT to initiate COAT reasoning format; (2) a large-scale RL finetuning stage to further enhance the self-reflection/exploration capabilities. For the SFT phase, a generator-critic-reward multi-agent data synthesis framework is used to construct high-quality demonstration trajectories. For the RL phase, a restart and exploration strategy is adopted, letting LLMs to reflect starting intermediate steps, and reward is carefully designed to have both rule-based correctness reward, reflection bonuses, and additional preference bonus. Extensive results demonstrate the effectiveness of COAT in reasoning tasks, in particular, the improvement over base models.

**Claims And Evidence:**

Yes, the claims in this work are made cautiously with supporting evidences.

**Essential References Not Discussed:**

NA

**Experimental Designs Or Analyses:**

I have checked the results provided in the main paper and the results are convincing.

**Methods And Evaluation Criteria:**

Yes, the proposed method is intuitively reasonable, a SFT phase to get initialized and a RL phase to further boost the performance. The evaluation is also comprehensive with in-domain and out-of-domain tasks with many other metrics presented.

**Other Comments Or Suggestions:**

NA.

**Other Strengths And Weaknesses:**

The paper is written very clearly, and it has been a pleasure to read it. The idea is nicely presented and the effectiveness is demonstrated with comprehensive experiments.

**Questions For Authors:**

One question that I would love to hear the author's opinion on: while this work considers a "more developed" framework to enhance models' self-improvement capability (e.g., use a SFT phase first to get COAT format and a large-scale RL phase), is it possible to compare with more naive approaches (e.g., the ones adopted in DeepSeek-R1 with only RL finetuning)? In particular, I would love to see both the performance comparison and the self-improvement pattern comparison.

I might have missed similar results in the paper. If that is the case, please provide a pointer.

**Relation To Broader Scientific Literature:**

The study of reasoning capability of LLMs are important to promote their usage in broader scientific domain, and I believe this work contribute to this field.

**Theoretical Claims:**

NA

---

> ### Author Rebuttal · Authors · 2025-03-31
>
> **Compare with more naive approaches (e.g., the ones adopted in DeepSeek-R1 with only RL finetuning).**
>
> We appreciate the reviewers' interest in the comparison between our method and the RL-only approach (r1-zero). As noted in our response to reviewer WTyn, r1 is a concurrent work, and such comparisons are not expected according to ICML reviewer guidelines. Nonetheless, we are happy to provide this comparison and share the observations and insights we have gained, in the spirit of contributing to the broader research community.
>
> The results and discussions can be found at https://docs.google.com/document/d/e/2PACX-1vRk1wLF9fsEVPFP2ijxNUd82LPnWZ2w-zTxVNGkkrof9yy36BNrM47JiCr6r8bga9c2Sr3q0a-S0oFF/pub
>
> * **Apple to apple performance comparison of r1-zero and Satori:**
>
> In order To fairly compare our method with r1-zero, we keep all experimental settings consistent, including the base model, training data, and RL training framework. The only differences lie in the algorithms used, our method employs PPO with RAE, while r1-zero uses GRPO, and whether an additional SFT stage is incorporated (ours includes Format Tuning and r1-zero starts RL training from the base model). **We evaluated r1-Zero on the same in-domain and out-of-domain benchmarks, and the results show no advantage over our method**. This suggests that **the perceived strong performance of the released DeepSeek-r1 model is likely due to significant engineering investment**, such as larger-scale in-house data, a much larger base model, and more powerful infrastructure. Moreover, **we observed several challenges and unexpected behavior of training r1-zero**, which we detail below.
>
> * **Training of r1-zero could be unstable:**
>
> We found that training r1-zero can be quite unstable. For instance, **rewards may drop sharply in later training stages, and the model sometimes begins generating repetitive random sequences**. This instability is especially common when the KL penalty is set to zero (i.e., no regularization). We hypothesize that without an SFT stage to anchor the model's behavior, the base model may produce harmful outputs early on, which can lead to a vicious cycle during optimization.
>
> * **R1-zero's behavior is hard to control:**
>
> Even when training converges, R1-Zero exhibits undesirable behaviors at inference time. Specifically, we observed two representative issues: (1) **Repetitive responses**, where the model generates repeated tokens or sentences mid-generation. (2) **Language mixing**, where responses may unexpectedly contain characters from other languages (e.g., Chinese), a phenomenon also noted by the r1 authors. These findings suggest that pure RL fine-tuning lacks model behavior constraints. While more carefully designed reward functions might help, we argue that a simpler and more effective solution is to incorporate a SFT stage, as we did with Format Tuning, to stabilize model’s initial behavior.
>
> * **R1-zero shows some "reflection pattern", especially using Python code verification:**
>
> We observe that r1-zero indeed demonstrates some reflection behavior after RL training, but we found **it tends to use Python code to verify answers, even when this is unnecessary**. For example, in a multiple-choice question from the MMLU-Pro dataset (which may not require logical verification), the model attempts to verify the answer using Python, which is a redundant step in that context. This behavior suggests that without appropriate constraints, **cold-start RL can introduce unexpected patterns, beyond the desired problem-solving skills**. Moreover, a recent study has also questioned the validity of the "Aha moment” claimed by r1-zero and suggests “Aha moment” could be a mirage [1].
>
> * **Our final takeaway: SFT warm ups, RL improves.**
>
> Our view is that **neither pure SFT nor pure RL (as in r1-Zero) is sufficient for effective LLM post-training**. Pure SFT can overly constrain the model and limit its ability to generalize beyond demonstrations. Pure RL can lead to unexpected and unreasonable behaviors, as observed with r1-zero. Instead, **we advocate for an appropriate combination of both**. This is the key insight we propose: a small-scale warm-up SFT helps the base model to get familiar with certain reasoning patterns, and a large-scale RL helps the model to utilize such reasoning patterns to incentivize its actual reasoning capability. This idea is also consistent with practices in the classical RL literature. For example, [2] demonstrates that a small amount of demonstration data can accelerate RL training by improving policy initialization. Moreover, the released DeepSeek-r1 model (as opposed to r1-Zero) also incorporates a SFT stage before RL (though the details of such SFT stage are not disclosed), which aligns with our insight.
>
> **Reference**
>
> - [1] There May Not be Aha Moment in R1-Zero-like Training — A Pilot Study, Feb, 2025.
> - [2] Deep Q-learning from Demonstrations, AAAI, 2018.

---

### Official Review · Reviewer_ZU1e · 2025-03-18

**Overall Recommendation:** 3

**Summary:**

In this paper, the authors proposes Satori as a framework for LLM reasoning. It is a two-stage framework, including Format Tuning and Self-improvement, to enhance LLM reasoning capabilities. The core contribution of this paper is the Chain-of-Action-Thought (COAT) framework, which structures LLM reasoning with meta-action tokens (<|reflect|> and <|explore|>) to enable self-correction and exploration. The Self-Improvement stage utilizes RL, the Restart and Explore (RAE) algorithm, to efficiently train the model to use the COAT format effectively. Experiments on math and out-of-domain benchmarks demonstrate that Satori outperforms existing models, exhibiting strong generalization and test-time scaling behavior.

**Claims And Evidence:**

Claims of superior performance and generalization are *partially* supported by empirical evidence.

* Strengths: Strong performance on math and out-of-domain benchmarks compared to baselines. Ablation studies show the contribution of reflection bonus and RL. Qualitative examples illustrate COAT reasoning.
* Weaknesses: Statistical significance is not explicitly discussed. Ablations are limited. Qualitative analysis is anecdotal. Benchmark scope is somewhat limited, and some benchmarks may be saturating. Lack of human-level performance comparison. Overall, while performance gains are shown, the strength and generality of the claims could be better supported with more rigorous and systematic evidence.

**Essential References Not Discussed:**

No essential references appear to be missing based on my current understanding of the literature.

**Experimental Designs Or Analyses:**

Experimental designs and analyses are generally sound, but could be strengthened.

* Strengths: Ablation studies provide some insights. Multi-agent data synthesis framework is a creative approach.
* Weaknesses: Ablations are limited in scope. Lack of statistical significance testing. Qualitative analysis is anecdotal. No error analysis of failure modes.

**Methods And Evaluation Criteria:**

Proposed methods (COAT + RAE) and evaluation criteria (math and reasoning benchmarks) are generally sensible for the problem.
* Strengths: COAT provides a structured approach to reasoning. RAE addresses RL challenges in reasoning. Benchmarks are relevant and challenging. Out-of-domain evaluation is a strong point.
* Weaknesses: Evaluation primarily focuses on final answer accuracy (pass@1). Metrics that evaluate reasoning process quality would be valuable. Benchmark scope is somewhat limited.

**Other Comments Or Suggestions:**

* Consider adding error analysis to understand common failure modes and how COAT helps address them.
* Minor comment: In Figure 1, "Trajectories Generation" could be rephrased to "Initial Trajectories Generation" for clarity.

**Other Strengths And Weaknesses:**

Strengths:
* Originality: Novel combination of COAT framework, RAE algorithm, and two-stage training for reasoning.
* Significance: Addresses an important problem (enhancing LLM reasoning) with a practical and efficient approach. Demonstrates promising empirical results and generalization.
* Clarity: Paper is generally well-written and easy to understand, especially with figures and examples.

Weaknesses:
* Incremental Novelty: Novelty is more in the combination than in fundamentally new concepts.
* Limited Rigor in Evaluation: Evidence could be strengthened with more systematic evaluations, statistical significance testing, and more detailed ablations.
* Justification of Design Choices: Some design choices (meta-actions, RAE parameters, reward function) could be more thoroughly justified and explored.

**Questions For Authors:**

1. Have you tried to ablate <|reflect|> and <|explore|> meta-actions separately? What if <|explore|> is not used at all?
2. Have you observed how COAT helps to mitigate the common failure modes? Which failure modes are best mitigated by COAT?

**Relation To Broader Scientific Literature:**

* Relation is clear: The paper clearly relates to the broader literature on LLM reasoning, CoT prompting, test-time search, self-improvement, and RL for LLMs.
* Specific relations: Builds upon CoT by adding meta-actions. Extends RL for reasoning by proposing RAE. Presents COAT as an alternative to data-intensive CoT fine-tuning and computationally expensive test-time search methods. Relates to SoS as concurrent work on training single LLMs for search, but argues for broader applicability of COAT.

**Theoretical Claims:**

This paper doesn't contain theoretical claims. No theoretical claims or proofs were checked.

---

> ### Author Rebuttal · Authors · 2025-03-31
>
> **1. Ablations and analysis are limited: Some design choices (meta-actions, RAE parameters, reward function) could be more thoroughly justified and explored.**
>
> We thank the reviewer for the suggestion. **We have conducted additional ablation studies to further analyze our design choices at** https://docs.google.com/document/d/e/2PACX-1vQ5lTnuQ5x6bx5Qh87cVdEq5iyrRIMLi5DYVHQUbrqB3f2Gye6mn0bHLwMcVosqddg_wp6P2JdEMfJM/pub.
> - Meta-actions: Please refer to rebuttal response 6.
> - RAE parameters: We clarify that RAE does not introduce any new tunable hyperparameters. However, to evaluate its impact, we perform an ablation study by training a model without applying RAE. The results show significant performance degradation, confirming that RAE is crucial for RL optimization.
> - Reward function: We have already ablated the reflection bonus in Appendix E. In addition, we include another ablation study that removes the preference bonus provided by the outcome reward model. The performance degrades in this setting, indicating that the preference bonus plays an important role in mitigating the sparse reward issues.
>
> **2. Analyze common failure modes and how COAT helps address them.**
>
> We thank the reviewer for the suggestion. **We have conducted a failure mode analysis (see provided link above)**. Specifically, we find that COAT effectively identifies and mitigates five common failure modes in reasoning tasks: (1) overly complicated solution that misleads the problem-solving process; (2) numeric simplification and calculations error; (3) replacing variables with incorrect numerical values; (4) lack of comprehensive consideration; (5) wrong interpretation of the problem.
>
> **3. Benchmark scope is limited.**
>
> Since our training data consists of open-sourced math problems, it is reasonable to evaluate the model primarily on math-related benchmarks. However, we would like to emphasize that **our evaluation goes beyond the math domain with six OOD datasets logical reasoning, code reasoning, commonsense reasoning, tabular reasoning, and domain-specific reasoning**. This type of OOD evaluation is rarely explored in prior works. We would appreciate it if the reviewer could clarify in what sense the benchmark scope is considered limited, and what types of additional evaluation would be most helpful.
>
> **4. Evaluation only focuses on final answer accuracy (pass@1).**
>
> We would like to note that zero-shot pass@1 is a widely adopted evaluation metric in LLM reasoning literature, provided that the ground-truth final answer is usually available. We agree with the reviewer that analyzing the intermediate reasoning process could provide valuable insights. However, this remains challenging due to the lack of reliable automatic verifiers capable of evaluating intermediate steps at scale. Toward a deeper understanding of our model’s behavior, **we have included a diverse set of demo examples in Appendix A that showcase different reasoning patterns of our model.**
>
>
> **5. Novelty is more in the combination than in fundamentally new concepts.**
>
> While the novelty is the eye of the beholder, we respectfully argue that the contributions of this work go beyond a simple combination of existing ideas. The reviewer describes our approach as a “novel combination of the COAT framework, RAE algorithm, and two-stage training for reasoning.” To the best of our knowledge, **none of these components have been introduced in prior LLM reasoning research**:
> * COAT differs substantially from classical CoT prompting in both design and motivations.
> * RAE offers a new perspective (by changing initial state distribution) to tackle the sparse reward problem in RL, a challenge has not been resolved by prior works.
> * Our two-stage training pipeline (small-scale Format Tuning + Online RL) also offers new insights for LLM post-training: SFT warm ups, RL improves.
>
> **6. Question-1: Have you tried to ablate <|reflect|> and <|explore|> meta-actions separately? What if <|explore|> is not used at all?**
>
> We believe that <|reflect|> and <|explore|> are fundamental meta-actions for reasoning tasks, and both are essential for effective problem-solving. Specifically, <|reflect|> prompts the model to evaluate its current reasoning, enabling it to identify potential errors or suboptimal steps, <|explore|> allows the model to propose alternative solutions, which is a natural follow-up when the model realizes (via <|reflect|>) that its current approach may be flawed.
>
> ***
> **If these clarifications satisfactorily address the reviewer's concerns, we kindly ask if the reviewer would consider updating the score to reflect what we believe is a paper with noteworthy contributions to the community.**

---

### Decision · Program_Chairs · 2025-05-01

**Decision:**

Accept (poster)

**Comment:**

In this work, the authors propose Satori, an LLM post training method which leverages Chain-of-Action-Thought (COAT) scheme to teach LLMs do reasoning. The overall pipeline has two key steps: 1) a small-scale SFT to teach LLM the COAT format; and 2) a larger-scale RL fine-tuning to teach the LLM self-reflection and exploration behaviors, via COAT reasoning format. The authors show that smaller models (e.g., Qwen 7b) trained with Satori can perform on par or exceed much larger LLM on a set of in distribution and out of domain benchmarks.

Reviewers agree that the multi-agent data synthesis framework is a neat idea; adding structure into reasoning paths (COAT), together with the Restart and Explore (RAE) algorithm, indeed seems to bring self-reflection and exploration capabilities to the model; the experiment results are strong, the out-of-domain results are especially promising.

Apart from a few clarification questions and asking for some additional ablation studies (which the authors did a good job addressing), a common concern raised by the reviewers is regarding the comparison with the DeepSeek R1 model (including R1-zero). This is mainly because Satori also aims to teach LLM models to do reasoning (and probably also involves some format training).

I am well aware that the R1 work was released one week before the ICML deadline (and one day before the abstract deadline), so I do not think it is necessary to compare with a concurrent work. I noticed both the authors and the reviewers that R1 is a concurrent work and thus I will not take such negative reviews into consideration.

During the rebuttal period, the authors nonetheless managed to conduct some analyses comparing their approach with R1/R1-zero, which is impressive and much appreciated. The additional analyses actually provided some interesting insights on the necessity of properly combine SFT and RL fine-tuning. Also given the fact that the authors promise to fully open-source Satori (code, data, and model), I feel this work can benefit the ICML community by providing a very interesting but different perspective on training reasoning models. Despite the majority of Satori's experiments only used a 7b base model, its strong performance gives me confidence that a similar approach could also work on a larger model.

After the rebuttal, one of the reviewers increased their score from 2 to 3, saying the framework is indeed a promising direction for future LLM training.

I recommend to accept this paper. Although the average score is not outstanding, if we ignored negative comments regarding R1 comparison, the score could be higher.